# Distinct stages in stress granule assembly and disassembly

Joshua R Wheeler[1†], Tyler Matheny[1†], Saumya Jain[1†], Robert Abrisch[1], Roy Parker[1,2*]

[1]Department of Chemistry and Biochemistry, University of Colorado Boulder, Boulder, United States; [2]Howard Hughes Medical Institute, University of Colorado, Boulder, United States

**Abstract** Stress granules are non-membrane bound RNA-protein (RNP) assemblies that form when translation initiation is limited and contain a biphasic structure with stable core structures surrounded by a less concentrated shell. The order of assembly and disassembly of these two structures remains unknown. Time course analysis of granule assembly suggests that core formation is an early event in granule assembly. Stress granule disassembly is also a stepwise process with shell dissipation followed by core clearance. Perturbations that alter liquid-liquid phase separations (LLPS) driven by intrinsically disordered protein regions (IDR) of RNA binding proteins in vitro have the opposite effect on stress granule assembly in vivo. Taken together, these observations argue that stress granules assemble through a multistep process initiated by stable assembly of untranslated mRNPs into core structures, which could provide sufficient high local concentrations to allow for a localized LLPS driven by IDRs on RNA binding proteins.

**\*For correspondence:** roy.parker@colorado.edu

[†]These authors contributed equally to this work

**Competing interests:** The authors declare that no competing interests exist.

## Introduction

Stress granules are non-membranous assemblies of mRNA and protein (mRNP) that form when translation initiation is limiting, which occurs during many stress responses. Stress granules are thought to influence mRNA function, localization, and to affect signaling pathways (*Buchan, 2014*; *Buchan and Parker, 2009*; *Kedersha et al., 2013*). Normally, stress granule formation is a dynamic, reversible process. However, pathological mutations in proteins that either increase the formation, or decrease the clearance of stress granules, can lead to abnormal accumulation of aggregates that share components with stress granules (*Buchan et al., 2013*; *Dormann et al., 2010*; *Izumi et al., 2015*; *Kim et al., 2013*). Abnormal accumulation of stress granule-like aggregates is associated with neurodegenerative disease (*Li et al., 2013*; *Ramaswami et al., 2013*). The molecular interactions and mechanisms that regulate stress granule assembly and how these may become altered in disease remain unknown.

Stress granules are members of an emerging class of non-membrane bound organelles and are thought to represent multicomponent viscous liquid droplets that from spontaneously by liquid-liquid phase separation (LLPS) (*Brangwynne, 2013*; *Brangwynne et al., 2009*, *2015*; *Elbaum-Garfinkle et al., 2015*; *Hyman and Simons, 2012*; *Hyman et al., 2014*). In vitro LLPS droplets display many properties of in vivo stress granules including fusion, wetting, shearing, and dynamicity (*Lin et al., 2015*; *Molliex et al., 2015*; *Murakami et al., 2015*; *Patel et al., 2015*; *Zhang et al., 2015*). LLPS in vitro can be driven by high local concentrations of proteins containing intrinsically disordered regions (IDRs) (*Lin et al., 2015*; *Molliex et al., 2015*; *Nott et al., 2015*; *Wright and Dyson, 2015*). Since IDR-containing proteins are enriched in stress granules and other mRNP granules (*Decker et al., 2007*; *Gilks et al., 2004*; *Jain et al., 2016*; *Kato et al., 2012*; *King et al., 2012*; *Reijns et al., 2008*), this has led to the suggestion that IDRs on mRNA binding proteins form hetero-

and homotypic interactions that drive initial formation of stress granules and P-bodies (*Gilks et al., 2004*; *Decker et al., 2007*; *Lin et al., 2015*; *Molliex et al., 2015*; *Patel et al., 2015*; *Toretsky and Wright, 2014*; *Weber and Brangwynne, 2012*; *Zhang et al., 2015*). This model is supported by observations in vitro wherein high concentrations of IDRs are sufficient to spontaneously form LLPS possibly through the interplay of weak electrostatic and hydrophobic homo- and heterotypic protein-protein interactions (*Lin et al., 2015*; *Molliex et al., 2015*; *Murakami et al., 2015*; *Nott et al., 2015*; *Pak et al., 2016*; *Patel et al., 2015*).

The interactions amongst IDRs driving phase-separation are enhanced by altering the molecular milieu either through lower salt concentration, molecular crowding, addition of RNA, or lower temperature (*Lin et al., 2015*; *Molliex et al., 2015*; *Nott et al., 2015*; *Patel et al., 2015*). In contrast, small aliphatic molecules, such as 1,6-Hexanediol, are hypothesized to perturb weak hydrophobic interactions, and thus disassemble assemblies that exhibit liquid-like properties in vitro (*Patel et al., 2007*; *Ribbeck and Görlich, 2002*). These in vitro observations suggest that manipulation of these properties might be useful for discerning the molecular interactions and mechanisms that drive assembly and regulate stress granules in vivo. Indeed, since mammalian stress granules have been reported to disassemble when cells are treated with 1,6-Hexanediol, they have been inferred to be LLPSs (*Kroschwald et al., 2015*).

Analysis of mammalian stress granules has revealed they are comprised of two phases: a less concentrated shell, which disassembles upon cell lysis and thus behaves like a LLPS formed by weak interactions; and more stable, internal 'core' structures (*Jain et al., 2016*). Interestingly, formation of in vitro liquid droplets by IDRs have been observed to mature into a second, less dynamic phase comprised in part of stable amyloid-like assemblies (*Han et al., 2012*; *Kato et al., 2012*; *Kwon et al., 2013*; *Lin et al., 2015*; *Molliex et al., 2015*; *Murakami et al., 2015*; *Patel et al., 2015*). A parsimonious model therefore is that stress granules may initially assemble to first form a LLPS comprised of multivalent, weak and dynamic interactions, and over time a less dynamic, more viscous second phase matures from the continued supersaturation of IDRs, thereby generating stress granule cores (*Jain et al., 2016*). Here, we set out to test several predictions of this model: (i) that lower temperature should enhance granule assembly, as LLPS driven by IDRs on RNA binding proteins have been shown to be enhanced at lower temperatures (*Molliex et al., 2015*; *Nott et al., 2015*); (ii) that stress granules are sensitive to drugs, such as 1,6-Hexanediol, that are predicted to disrupt weak hydrophobic interactions; (iii) that stress granule dynamics decrease and core size should increase over time as stress granule structure matures.

In contrast to the expected results, we observed that stress granule assembly is inhibited at lower temperatures, despite the extent of translation repression being temperature independent. Moreover, we observe that 1,6-Hexanediol triggers stress granule formation in both yeast and mammalian cells, as well as altering many cellular structures including the cytoskeleton and the nuclear pore complex. Finally, we observe stress granule dynamics and core size is unchanged during prolonged stress. Given these observations we suggest an alternative model where untranslating mRNPs initially oligomerize into stable cores, thereby nucleating an initial shell layer and providing a platform for the growth of a more dynamic shell around these cores, followed by merger of these individual core/shell assemblies into large mature stress granules. Stress granule disassembly is also a stepwise process where shell dissipation occurs first followed by clearance of cores.

## Results

### Two models of granule assembly can explain a biphasic stress granule architecture

We have previously described that mammalian stress granules formed after 60 min of sodium arsenite ($NaAsO_2$) treatment contain substructures (referred to as cores) that are stable in lysates, as well as a less concentrated surrounding structure, referred to as the shell (*Jain et al., 2016*). The shell region of stress granules does not persist in our lysis conditions, which suggests it is sensitive to dilution, one of the properties of an LLPS.

The biphasic stress granules seen after 60 min in mammalian cells could form by one of two models (*Figure 1A*). First, the increased pool of untranslated mRNAs bound by proteins containing IDRs, could lead to the formation of a LLPS based on IDR-IDR interactions that would be very fluid and

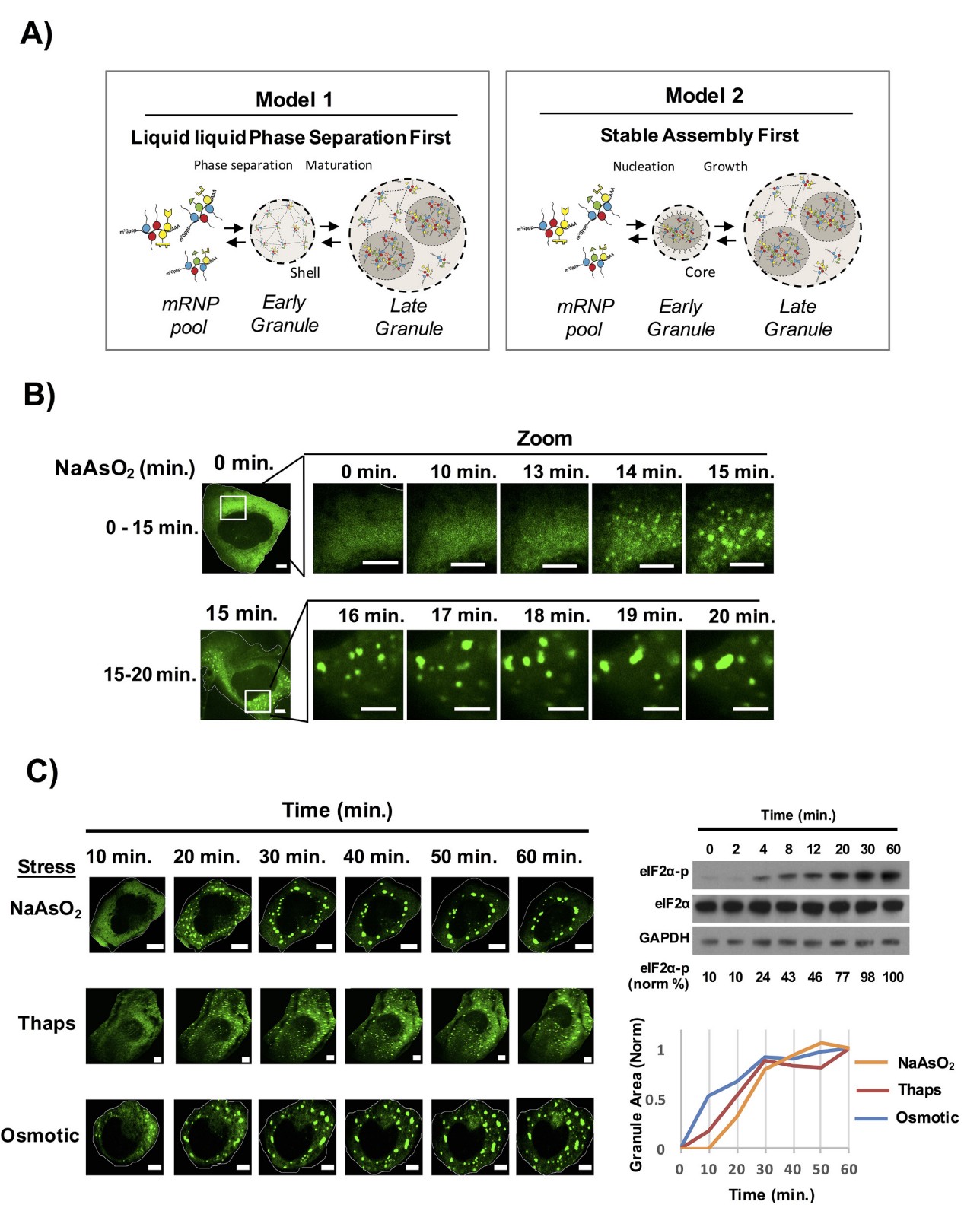

**Figure 1.** Time course of stress granule assembly. (**A**) Two models for stress granule assembly. In Model 1, an increase in a pool of untranslated mRNAs with bound proteins containing IDRs, could lead to the formation of a LLPS based on weak, dynamic IDR-IDR interactions and over time core formation occurs as a second phase due to the supersaturation of local concentration of core components. In Model 2, untranslating mRNAs with bound proteins containing IDRs could initially oligomerize into stable cores that provide a platform for LLPS and eventual coalescence of multiple cores results in

*Figure 1 continued*

formation of a larger LLPS assembly. (**B**) Time-lapse imaging stress granule assembly and early dynamics in U-2 OS cells expressing GFP-G3BP1 during NaAsO$_2$ stress using 100X objective. (**C**) Time-lapse imaging of stress granule assembly under NaAsO$_2$ (0.5 mM), thapsigarin (Thaps, 100 nM), or osmotic stress (375 µM sorbitol). Western blot for eiF2α phosphorylation status following exposure to NaAsO$_2$ stress over time. Percent eiF2α phosphorylation was normalized to total eiF2α. eiF2α and GAPDH serve as loading controls. Graph shows average granule area plotted for each stress condition normalized to maximal granule area at 60 min. All scale bars are 5 µm.

dynamic. In this view, cores would assemble within this initial LLPS in a second step, possibly due to the increased local concentration of core components (*Jain et al., 2016*). In support of this view, LLPS derived from RNA binding protein IDRs in vitro mature to include more stable, possibly amyloid-like, substructure (*Lin et al., 2015*; *Molliex et al., 2015*; *Murakami et al., 2015*; *Patel et al., 2015*).

In a second model, stress granule formation could be initiated by the oligomerization of mRNPs into stable core structures. Because these stable core structures would concentrate mRNA binding proteins with IDRs, stable cores could function as nucleation sites for LLPS by creating a high local concentration of IDRs to drive LLPS formation. Over time, cores may then form into larger LLPS assemblies with multiple core structures within each stress granule. In this latter model the formation of the 'shell' structure of the stress granule would be analogous to the formation of the nuclear pore, wherein a high local concentration of FG-repeat containing proteins is imposed by the structure of the nuclear pore, which then facilitates the formation of phase transition in the pore region (*Frey and Görlich, 2007*; *Hülsmann et al., 2012*; *Schmidt and Görlich, 2016*).

To distinguish between these two models for stress granule assembly we examined the time course of stress granule assembly, as well as the biochemical, ultrastructural, and dynamic properties of stress granules at different stages of their assembly.

To examine the time course of stress granule assembly, we exposed human osteosarcoma cells (U-2 OS) expressing a GFP-G3BP1 fusion protein (*Kedersha et al., 2008*; *Ohn et al., 2008*) to NaAsO$_2$ and collected images over time using a spinning disc confocal microscope. We observed that stress granules began to form 13–15 min after stress induction, which correlated with an increase in eIF2α phosphorylation (*Figure 1B,C*, *Video 1*). The initial stress granules then grew in size both by accretion of material and by fusion with other stress granules (*Figure 1B*). The area of GFP-G3BP1 stress granules continued to increase with the majority of stress granule assembly completed by 40 min. Stress granules induced with thapsigargin or osmotic stress showed a similar time course of stress granule growth albeit the rates of granule induction were slightly different (*Figure 1C*). Our data is consistent with the idea that assembly of a mature stress granule occurs via a multistep process: an initial nucleation event that may be influenced by the stress stimulus and a consistent growth phase that requires an addition of substrate and fusion of the small initial stress granules into larger assemblies.

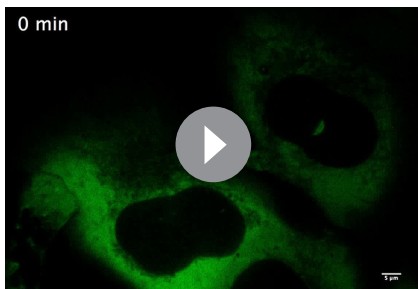

**Video 1.** Stress granule assembly during NaAsO$_2$ stress. U-2 OS cells expressing GFP-G3BP1 were exposed to NaAsO$_2$ (0.5 mM) and imaged for 20 min. Images were acquired at 20 s intervals on a spinning disk confocal microscope using a 100X objective. Scale bar represents 5 µm. Related to *Figure 1*.

## Core formation is an early step in stress granule formation

If a stress granule shell forms first, then stress granules formed at early time points should be comprised of only a shell, no core structures, and would not be stable in lysates. To test this, we examined if the earliest detectable stress granules contain compartments that are stable in lysates (*Jain et al., 2016*).

Lysis of cells at 5 min, 10 min, and 15 min after NaAsO$_2$ induction revealed that by 15 min, when stress granules are first visible in cells, we observed an increase in the number of stable stress granule cores detected in lysates (*Figure 2A*). We identify these GFP-G3BP1 foci in lysates as stress granule cores since they

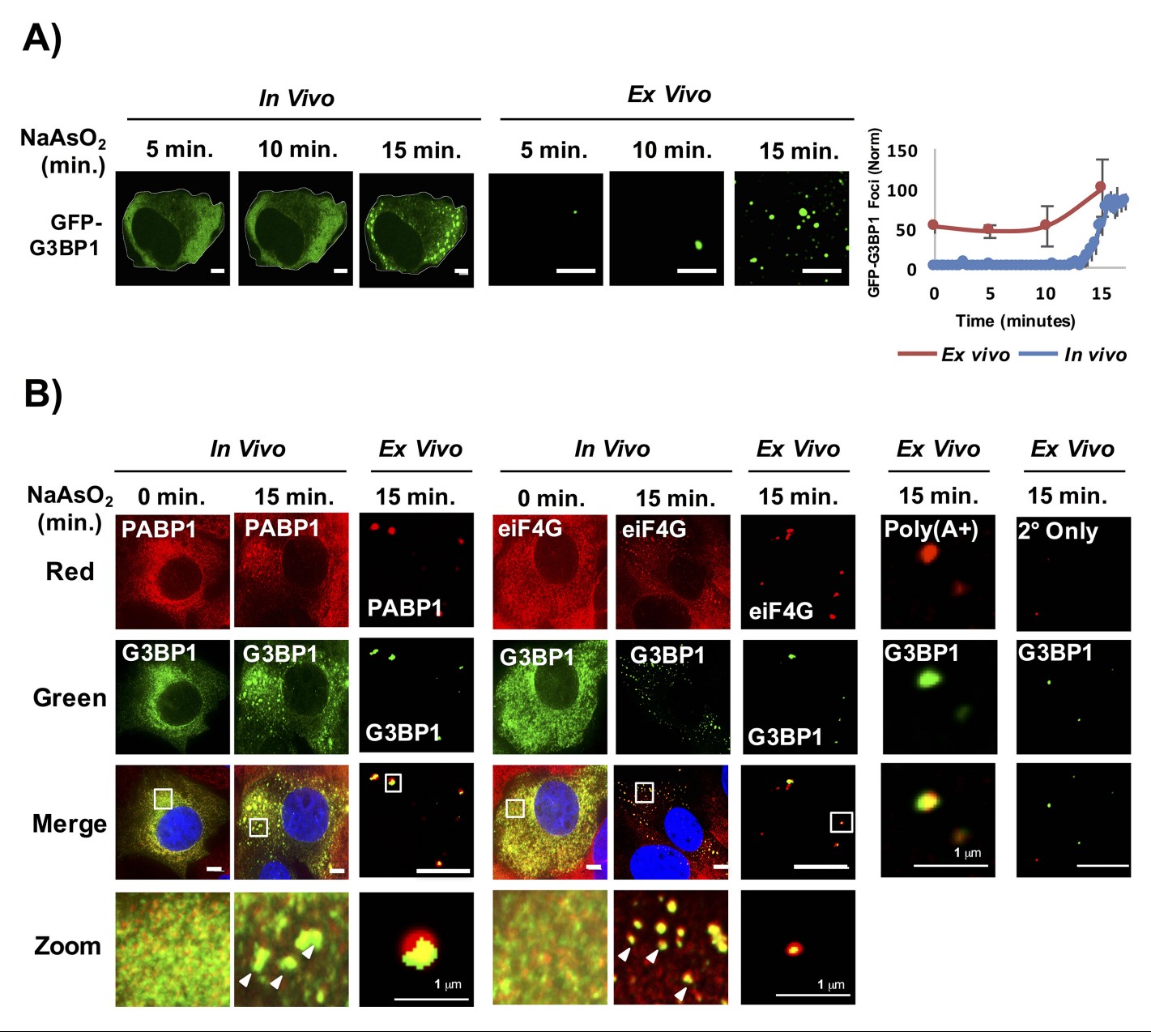

**Figure 2.** Stress granule cores are stable in lysates early in granule assembly. (**A**) GFP-G3BP1 U-2 OS cells and cell lysates following treatment with NaAsO₂ for 15 min. Graph shows percent GFP-G3BP1 foci for cells (in vivo) and cell lysates (in vitro) and normalized to maximal number of foci detected at 15 min (in vivo) or number of foci detected at time 0 min (in vitro). (**B**) GFP-G3BP1 U-2 OS cells and cell lysates probed for the stress granule markers, PABP1 or eIF4G following treatment with NaAsO₂ for 15 min. GFP-G3BP1 foci in lysates probed for poly (A+) RNA by oligo-dT or secondary-only antibody from cells NaAsO₂ stressed for 15 min. GFP-G3BP1 foci in lysates were probed with secondary antibody only (Alexa-647) at same concentrations used for primary (PABP1 or eIF4G) antibody detection. Zoom represents magnified inset. Unless otherwise noted, all scale bars are 5 μm.

contain PABP1, poly(A)+ RNA, and eIF4G (*Figure 2B*). Similarly, yeast stress granules are stable in lysates as soon as they are microscopically detectible in vivo (data not shown). We interpret this observation to indicate that as soon as stress granules are observed in cells by microscopy they contain core assemblies that are stable in lysates.

## Early and late stress granules are non-uniform by super resolution microscopy

As an additional test to see if stress granules formed at early times contain core substructures, we examined the ultrastructure of stress granules by super-resolution microscopy (Structured Illumination Microscopy [SIM]) at different times. Using SIM for G3BP1, as assessed by mapping pixel intensity for individual granules, we observed that stress granules contained substructure as soon as they were large enough to be examined by SIM (20 min) (*Figure 3*).

Based on these two observations, we conclude that core assembly represents an early event in stress granule formation.

## Stress granule core size and dynamics do not change over time

Previous in vitro experiments demonstrate that IDR-containing stress granule proteins can undergo a maturation event leading to the formation of more stable complexes and possibly amyloid-like assemblies (*Kato et al., 2012*; *Lin et al., 2015*; *Molliex et al., 2015*; *Murakami et al., 2015*; *Nott et al., 2015*; *Patel et al., 2015*; *Zhang et al., 2015*). To test if stress granules show a similar 'maturation' in cells, we examined both the dynamics and core dimensions at different times during stress granule formation.

We examined the properties of the stable components of stress granules in lysates to see if there was an increase in size of the core structures over time, which would imply a continued growth of these stable core structures and might reveal maturation of the core structure within a larger stress granule. By Nanosight Nanoparticle Tracking analysis of lysates from U-2 OS cells at different times after stress, we observed that GFP-G3BP1 granule core size does not significantly change between 15 min (median size, $259 \pm 74$ nm), 30 min (median size, $258 \pm 24$ nm), 60 min (median size, $273 \pm 12$ nm), and 120 min (median size, $247 \pm 4$ nm) of $NaAsO_2$ stress (one-way ANOVA, p-value = 0.94)

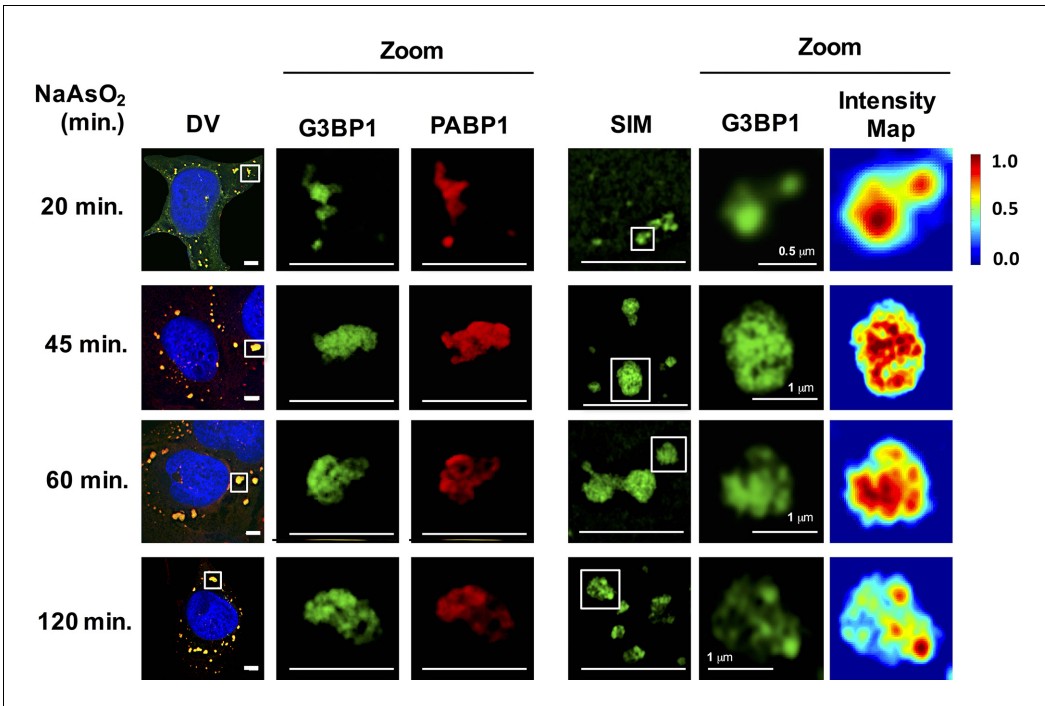

**Figure 3.** Early and late stress granules contain a non-uniform substructure. GFP-G3BP1 stress granules were stained with the stress granule marker PABP1 and imaged by deconvolution microscopy (DV) following $NaAsO_2$ stress at different time points. GFP-G3BP1 was assessed in the same cells by structured illumination microscopy (SIM). Intensity map represents relative gray scale intensity. Zoom represents magnified inset. Unless otherwise noted, all scale bars are 5 µm.

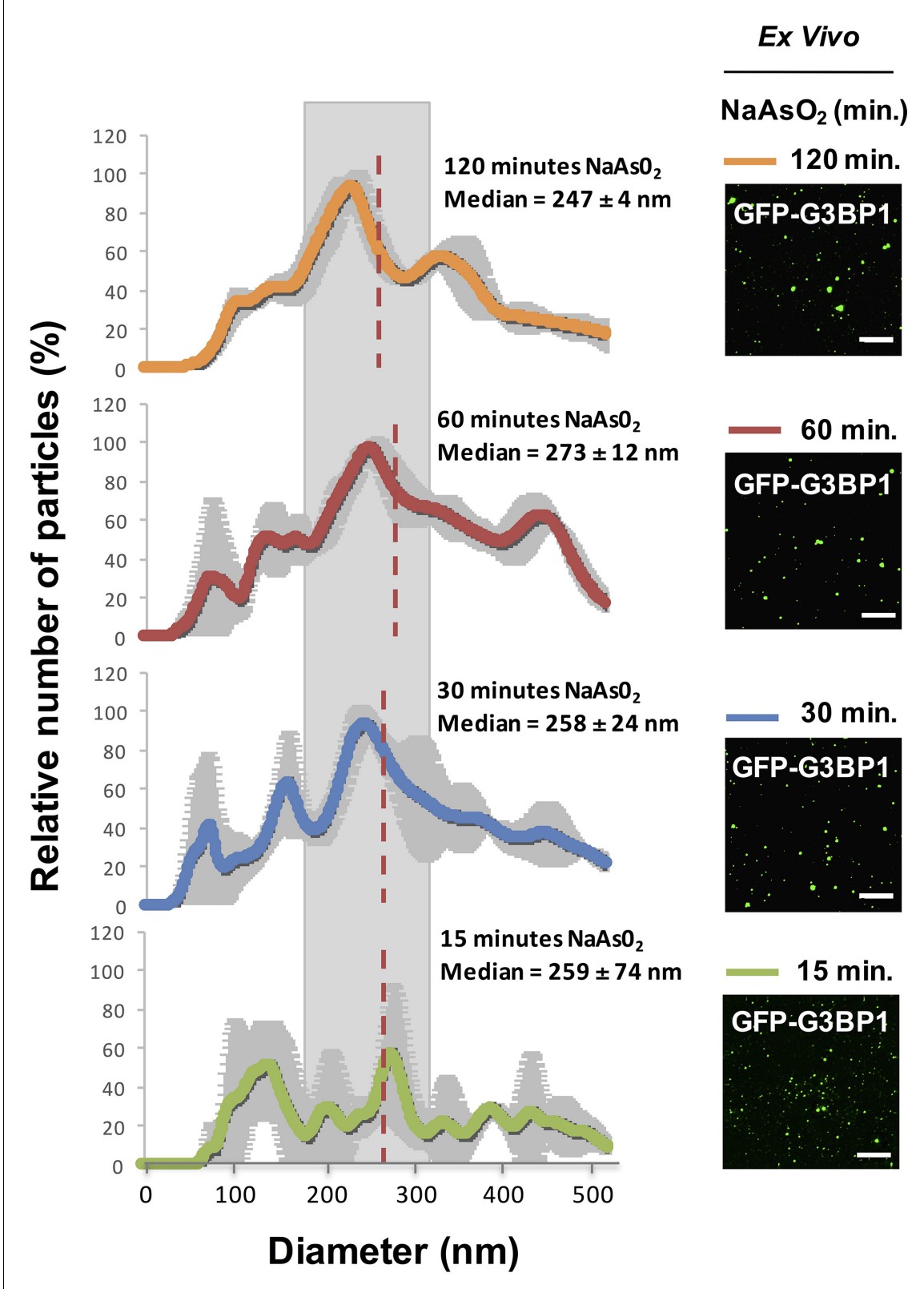

**Figure 4.** Similar GFP-G3BP1 stress granule core size during early and prolonged stress. Mean of 3 Nanosight experiments ± standard deviation for GFP-G3BP1 mammalian stress granule cores isolated from U-2 OS cells stressed with $NaAsO_2$ for 15, 30, 60, or 120 min. Relative number of particles for each time point are represented as a percentage of maximal number of particles per unit size. Mean median from these 3 experiments is highlighted. Representative images from GFP-G3BP1 cores in lysates at respective time points are shown. All scale bars are 5 µm.

(*Figure 4*). We interpret these results to demonstrate that GFP-G3BP1 granule core size does not appreciably mature into larger stable assemblies within the time periods assessed.

In a second experiment, we examined if the dynamics of stress granules change over time as assessed by FRAP (Fluorescence Recovery After Photobleaching) analysis of GFP-G3BP1. This experiment is analogous to experiments done in vitro where the recovery rates of LLPS driven by IDRs of RNA binding components decrease over time revealing a maturation process within these reconstituted assemblies (*Lin et al., 2015*; *Molliex et al., 2015*; *Murakami et al., 2015*; *Patel et al., 2015*; *Zhang et al., 2015*). Thus, we examined if the dynamics of normal stress granules change over time, which might reveal an analogous maturation process in cells. We observed that the recovery rates of GFP-G3BP1 were essentially identical after 30 min, 60 min, or 120 min of exposure to $NaAsO_2$ (*Figure 5*), although the half-time of recovery was slightly faster at 30 min (18.5 s) as compared to 60 min (35 s) and 120 min (31 s). This slightly faster recovery rate is likely to simply reflect that at 30 min stress granules are still undergoing growth, which will increase the apparent rate of recovery. We interpret this observation to argue there is not a global transition of stress granule components to a more stable state within the time periods assessed. However, we cannot rule out that specific components of stress granules become less dynamic over time.

## Mammalian stress granule assembly is inhibited at lower temperatures

Our observations suggest that core formation is an early event in stress granule assembly; however, we cannot distinguish by light microscopy if cores precede shell formation or not. To distinguish between these early events, we first tested whether stress granule assembly is enhanced at lower temperatures. The logic of this experiment is based on multiple experiments in vitro showing that IDRs from RNA binding proteins can form either LLPS (*Lin et al., 2015*; *Molliex et al., 2015*; *Murakami et al., 2015*; *Nott et al., 2015*; *Zhang et al., 2015*) or hydrogels, which in this context are a meshwork assembly of protein filaments (*Han et al., 2012*; *Kato et al., 2012*). In these

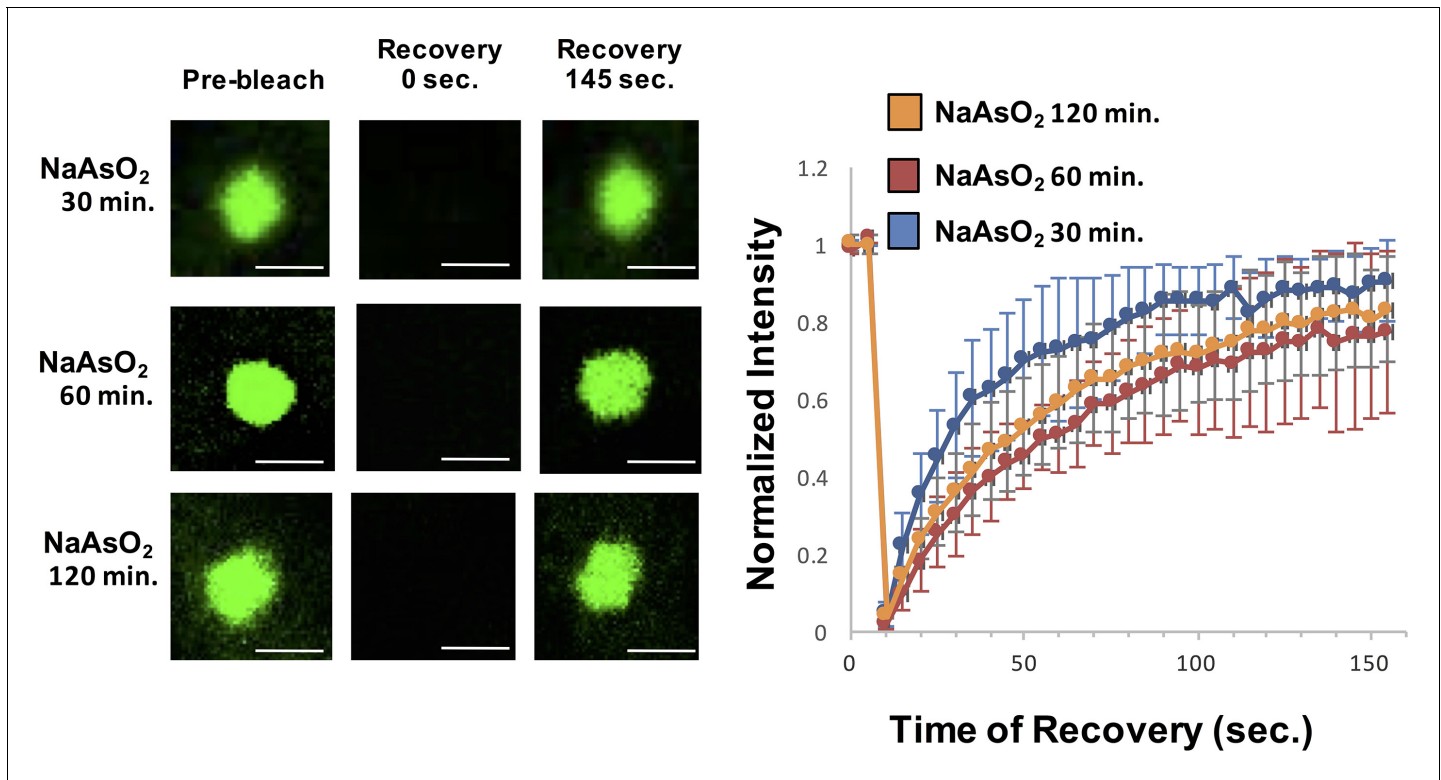

**Figure 5.** Similar GFP-G3BP1 stress granule dynamics during early and prolonged stress. Granules shown prior to photobleaching, at 0 s, and at 145 s after photobleaching. Cells were treated for either 30, 60, 120 min with $NaAsO_2$. Graph shows recovery curves as an average of 6 granules ± standard deviation at each respective time point. Scale bars are 1.5 μm.

experiments, the LLPS or hydrogel formation is enhanced at lower temperatures, which would be expected to lower the critical concentration required for assembly (*Molliex et al., 2015*; *Murakami et al., 2015*; *Nott et al., 2015*; *Kato et al., 2012*). If these types of interactions drive stress granule assembly in vivo, then stress granule assembly should be more efficient at lower temperatures. To test this possibility, we examined stress granule formation at different temperatures in GFP-G3BP1 U-2 OS cells, in response to NaAsO$_2$ treatment.

We observed that stress granules formed less efficiently at lower temperatures with a continuing decline in the rate, overall number and size of stress granule formed as the temperature declined from 37°C to 27°C (*Figure 6A,B*). Two observations suggest that lower temperatures do not perturb translational inhibition. First, we observe a stress-specific induction of eiF2α-phosphorylation to similar levels amongst all the temperatures assessed (*Figure 6C*). Second, via polysome profiling, we observed a similar extent of translation repression after 20 min of NaAsO$_2$ stress at 30°C as compared to 37°C. (*Figure 6E*). Taken together, we conclude that despite a sufficient extent of translation repression and an environment shown to be conducive for phase separation in vitro (i.e. lower temperatures), we observe a delay in stress granule formation at lower temperatures.

In principle, stress granule formation could be dependent on higher temperatures either because the interactions holding stress granules together are disrupted at lower temperatures, or because the assembly of stress granules is an ordered process with some step in the process dependent on higher temperatures. If the interactions holding stress granules together are dependent on higher temperatures, then stress granules formed at higher temperatures should disassemble at lower temperatures. Conversely, if stress granules are assemblies that have a temperature dependent step in their assembly, then once formed they should persist after a drop in the temperature. To test these possibilities, we induced stress granule formation at 37°C for 45 min then imaged the cells at a lower temperature (30°C) for 15 min (*Figure 6D*). We observed that once formed, stress granules persist at a lower temperature and do not appreciably decrease in size. It should be noted that shifting cells to 30°C (without NaAsO$_2$) does not induce stress granule formation (*Figure 6D*). This is inconsistent with stress granules being held together by interactions that are enhanced at higher temperatures, and instead argues stress granules have a step in their assembly that is inhibited at low temperatures, but once formed stress granules are stable at both high and low temperatures. This suggests that the primary driving forces of stress granule assembly are not weak interactions between IDR domains that are enhanced at low temperatures.

## 1,6-Hexanediol disrupts cellular structures in yeast and alters cellular morphology in HeLa cells

Our observations suggest that promoting LLPS by lower temperature is not sufficient to enhance granule formation. To test if an LLPS shell precedes and is necessary for granule core, we decided to test the effect of inhibiting or disrupting LLPS formation by the addition of 1,6-Hexanediol.

1,6-Hexanediol has previously been used to disrupt various structures that are expected to represent LLPS, in vitro and in vivo (*Kroschwald et al., 2015*; *Molliex et al., 2015*; *Patel et al., 2007*; *Ribbeck and Görlich, 2002*; *Updike et al., 2011*). As the nuclear pore is proposed to be a LLPS (*Patel et al., 2007*; *Ribbeck and Görlich, 2002*; *Updike et al., 2011*), a prediction is that 1,6-Hexanediol should disrupt this structure in vivo. Consistent with this prediction, the addition of 10% 1,6-Hexanediol to yeast cells (*Kroschwald et al., 2015*), rapidly disrupted the nuclear pore, as adjudged by the loss of punctate, peri-nuclear localization of the nuclear pore protein, Nsp1 (*Figure 7A*). However, we noticed that the effect of 1,6-Hexanediol was not limited to cellular structures that are known to represent a LLPS. We observed that 1,6-Hexanediol (but not 1, 4, 6-Hexanetriol) caused a significant disruption of actin and tubulin organization, as adjudged by the localization of Sac6-GFP (an actin binding protein) and Tub1-GFP (Alpha-Tubulin) respectively (*Figure 7A* and data not shown). This disruption of cytoskeleton organization occurred within two minutes of 1,6-Hexanediol treatment. Interestingly, Sac6 changed localization twice – once rapidly to go from punctate to diffuse; and later from diffuse to into rod-like structures in the cytoplasm (*Figure 7A*). Membrane bound cellular structures such as the Endoplasmic Reticulum (ER, observed using Get1-GFP) and mitochondria (observed using Atp1-GFP) were not affected by 1,6-Hexanediol. These observations suggest that 1,6-Hexanediol will disrupt a variety of cellular structures.

We also examined how 1,6-Hexanediol affected mammalian cells. Two observations suggest 1,6-Hexanediol affects cellular viability and gross cellular morphology of mammalian cells. First, HeLa

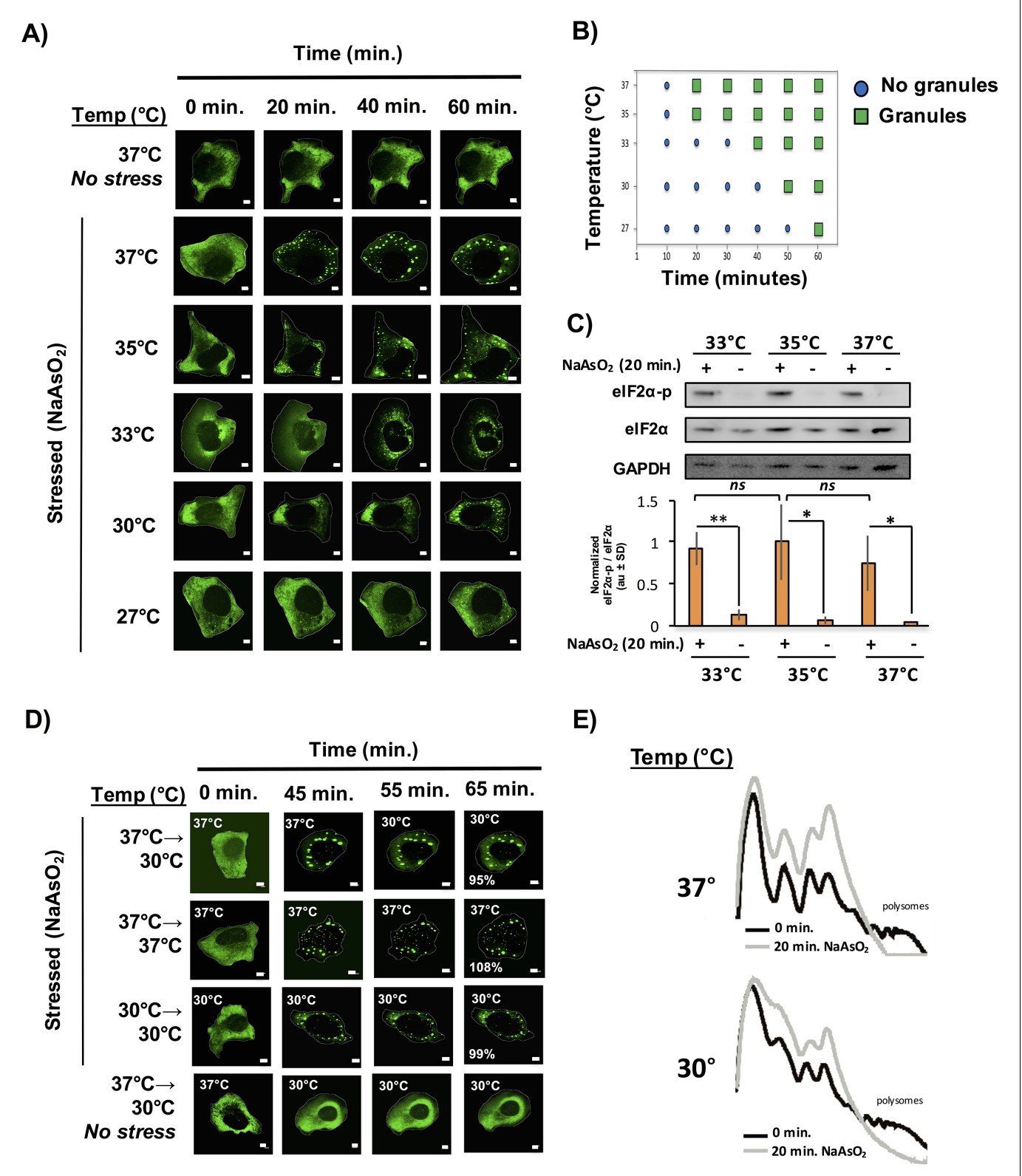

**Figure 6.** Temperature inversely affects kinetics of stress granule formation. (**A**) Time-lapse microscopy of U-2 OS cells expressing GFP-G3BP1. U-2 OS cells imaged at respective temperatures in the presence of $NaAsO_2$ stress for 1 hr using 100X objective. (**B**) Graphical representation of temperature on stress granule formation in U-2 OS cells. (**C**) Western blot for eiF2α phosphorylation status following +/− exposure to 20 min of $NaAsO_2$ stress at respective temperatures. eiF2α and GAPDH serve as loading controls. Normalized to total eiF2α (N=3, au = arbitrary units, SD = standard deviation). (p-

*Figure 6 continued*

value: *<0.05; **<0.01; ns = not significant). (**D**) Time-lapse microscopy of U-2 OS cells expressing GFP-G3BP1 at respective temperatures following 45 min NaAsO$_2$ stress using 100X objective. Temperatures at which cells were imaged are indicated in top left of each panel. Percentages listed in final panel represent average area of granules normalized to the area of granules at the start of image acquisition. (**E**) Polysome analysis of U-2 OS cells (grown either at 30°C or 37°C) during steady state or following 20 min NaAsO$_2$ stress. All scale bars are 5 μm.

cells demonstrated a 40% reduction in cellular viability as assessed by RealTime-Glo MT Cell Viability Assay following exposure to 3.5% 1,6-Hexanediol as compared to untreated cells. Second, HeLa cells exposed to 1,6-Hexanediol demonstrated rapid widespread morphological changes within seconds including membrane blebbing (*Figure 7B*; *Video 2*). Thus, owing to the adverse effects of 1,6-Hexanediol on cell viability and morphology, it's use in mammalian cells in culture may be complicated.

## 1,6-Hexanediol induces stress granule formation in yeast

To test the effect of 1,6-Hexanediol on already assembled stress granules and P-bodies in yeast, we glucose starved yeast cells for 15 min, followed by incubation with 10% 1,6-Hexanediol as previously described (*Kroschwald et al. 2015*). Markers of stress granules and P-bodies changed localization twice (similar to Sac6-GFP above). First, both assemblies quickly disassembled within two minutes, however, after 10 min, they reappeared, often completely co-localized (*Figure 8A*). This is in contrast to a previous report that suggested that yeast P-bodies, but not stress granules, are sensitive to 1,6-Hexanediol treatment under glucose starvation conditions (*Kroschwald et al. 2015*). We observed that P-bodies formed upon 1,6-Hexanediol treatment are less intense than the P-bodies formed under glucose starvation conditions and this may have limited their earlier detection.

By using strains with tagged components of stress granules and P-bodies, we observed that in the absence of any other stress, 1,6-Hexanediol (and not a Digitonin control) also induced robust Pab1-GFP (a stress granule marker) granule formation within 10 min of treatment (*Figure 8B*). Interestingly, these Pab1-GFP foci also overlapped with Edc3-mCherry – a P-body protein (*Figure 8B*). Thus, exposure of yeast cells to 1,6-Hexanediol is an effective inducer of stress granule-like assemblies.

## 1,6-Hexanediol induces stress granule formation in HeLa cells

To determine how pre-existing mammalian stress granules react to 1,6-Hexanediol, we induced stress granules with NaAsO$_2$ in HeLa cells and then treated cells with 1,6-Hexanediol. Similar to yeast stress granules, we observed an initial reduction in total stress granule numbers, which coincided with alterations in cellular morphology; however, after 10 min, stress granules re-appeared (*Figure 9A*). These effects of 1,6-Hexanediol were also observed in U-2 OS cells, suggesting this is not a cell specific phenotype (data not shown).

In the absence of any other stress, 1,6-Hexanediol treatment resulted in the formation of stress granule-like assemblies. These assemblies stained positive for stress granule markers G3BP1 and PABP1 (*Figure 9B*). Thus, similar to yeast cells, 1,6-Hexanediol treatment of mammalian cells in culture can induce stress granule-like assemblies.

Several observations suggest the assemblies induced by 1,6-Hexanediol are bona fide stress granules. First, 1,6-Hexanediol induces the formation of assemblies that contain known stress granule proteins in both yeast and mammalian cells (HeLa and U-2 OS cells) (*Figure 10A*). Second, 1,6-Hexanediol induced assemblies are sensitive to cycloheximide treatment, a drug known to block stress granule assembly, in both yeast and mammalian cells (*Figure 10A*). Third, 1,6-Hexanediol-induced assemblies are stable in lysates suggesting that the material properties of these assemblies contain a relatively stable set of interactions similar to normal stress granules (*Figure 10B*). Fourth, eiF2α phosphorylation is induced to a similar extent by 1,6-Hexanediol compared to NaAsO$_2$ (*Figure 10C*). Finally, FRAP analysis of 1,6-Hexanediol-induced granules in mammalian cells indicates that 1,6-Hexanediol induced stress granules behave similar to normal stress granules (*Figure 10D*).

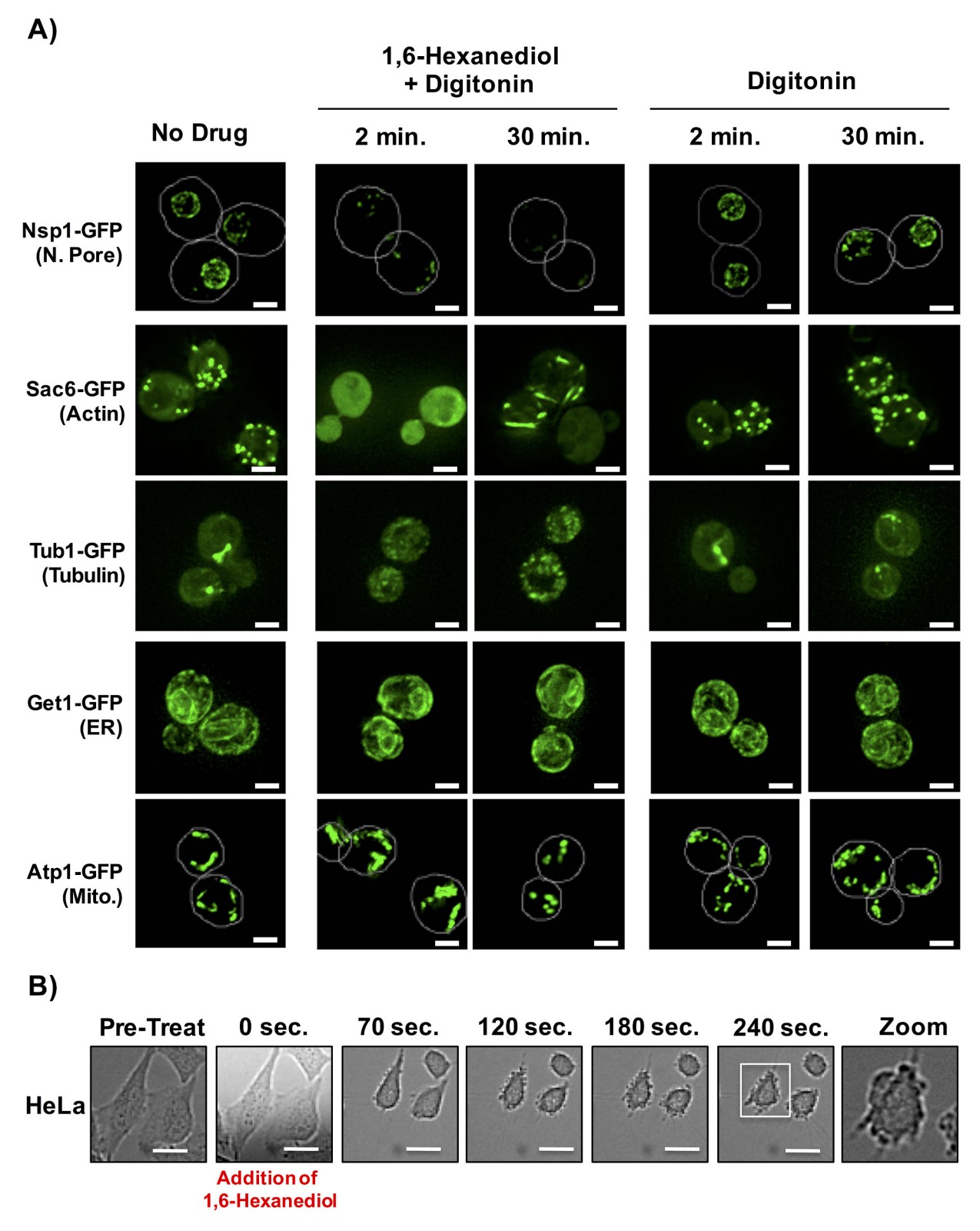

**Figure 7.** 1,6-Hexanediol disrupts many cellular structures in yeast and alters cellular morphology in HeLa cells. (**A**) The localization of various GFP-tagged proteins is shown either without treatment (No Drug); in the presence of 10% 1,6-Hexanediol and 10 µg/mL Digitonin for 2 min or 30 min; or just 10 µg/mL Digitonin for 2 min or 30 min. For Nsp1-GFP and Atp1-GFP, white lines depict cell boundaries. N. Pore = nuclear pore. ER = Endoplasmic Reticulum. Mito. = Mitochondria. Get1, Guided Entry of Tail-anchored proteins (YGL020C). Atp1, ATP synthase (YBL099W). Scale bar is 2

*Figure 7 continued on next page*

*Figure 7 continued*

μm. (**B**) Time-lapse microscopy of HeLa cells treated with 3.5% 1,6-Hexanediol reveals alterations in cellular morphology. HeLa cells were exposed to 3.5% 1,6-Hexanediol (time zero) and imaged for 5 min using a 20X objective. Scale bars are 20 μm.

## Stress granules disassemble through multiple discrete steps

Our data suggests a model for stress granule assembly wherein mRNPs oligomerize to form stable cores and over time individual cores then dock with one another through a more dynamic shell. We reasoned stress granule disassembly may occur in a reverse process where a less stable shell may dissipate initially followed by core disassembly or clearance. To test this prediction, we examined stress granule disassembly using live cell imaging. Several observations suggest that, like stress granule assembly, stress granule disassembly occurs in a multi-step process.

First, live cell imaging of stress granule disassembly shows that stress granules start out as large complexes and disassemble within a narrow time window. Within the limitations of our ability to detect stress granule disassembly, we observe for large stress granules, disassembly appears to occur in two steps: first, larger stress granules break into smaller foci followed by disassembly and/or clearance of these smaller foci. During disassembly, we observe the appearance of filamentous structures emerging at the edges of the disassembling stress granule; however, we should point out that due to the resolution of our microscope we are unable to characterize the nature of these structures (*Figure 11BC*). Interestingly, once stress granules disassemble into smaller assemblies, these small assemblies become microscopically undetectable within a similar time scale as required for their initial detection during assembly (*Figure 11A,B*, *Video 3*).

Second, examination of stress granule disassembly reveals non-uniform distribution within stress granules undergoing disassembly wherein areas of relative concentration appear to persist once the surrounding shell structures have dissolved (*Figure 11B*). Finally, we observe stress granules are stable in lysates at the same time points that disassembly is occurring (*Figure 11C*).

Taken together, stress granule disassembly may occur through multiple steps wherein RNA is titrated out of stress granule into translation leading to structural instability and subsequent disassembly of a larger stress granule complex into smaller core structures that are then disassembled or cleared by autophagy.

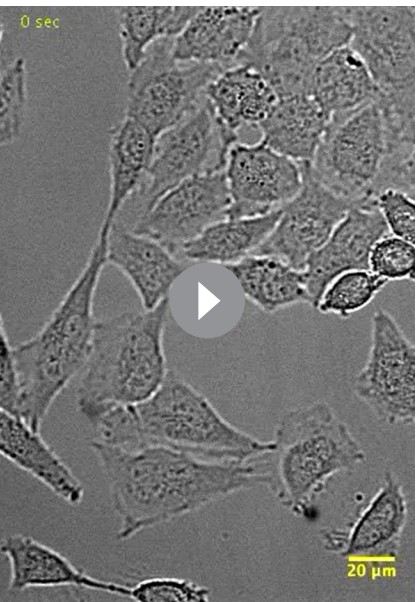

**Video 2.** HeLa cells exposed to 1,6-Hexanediol. HeLa cells were exposed to 1,6-Hexanediol (3.5%) and imaged for 5 min. Images were acquired at 10 s intervals on a spinning disk confocal using a 20X objective. Scale bar represents 20 μm. Related to *Figure 7*.

## Discussion

In this manuscript, we set out to distinguish between two different models to explain the processes of stress granule formation, maturation, and disassembly. In the first model, stress granules form by a loose condensation of IDR containing proteins to form an initial phase-separated, dynamic structure and over time stable cores mature within this dynamic structure. This model is suggested by recent findings in vitro, demonstrating LLPS can be driven by IDRs and these LLPS subsequently mature to form a second, less dynamic phase (*Lin et al., 2015*; *Molliex et al., 2015*; *Murakami et al., 2015*; *Patel et al., 2015*; *Zhang et al., 2015*). In the second model, we considered that stress granules initially condense to form stable cores, possibly with a nascent shell layer, and these cores merge into

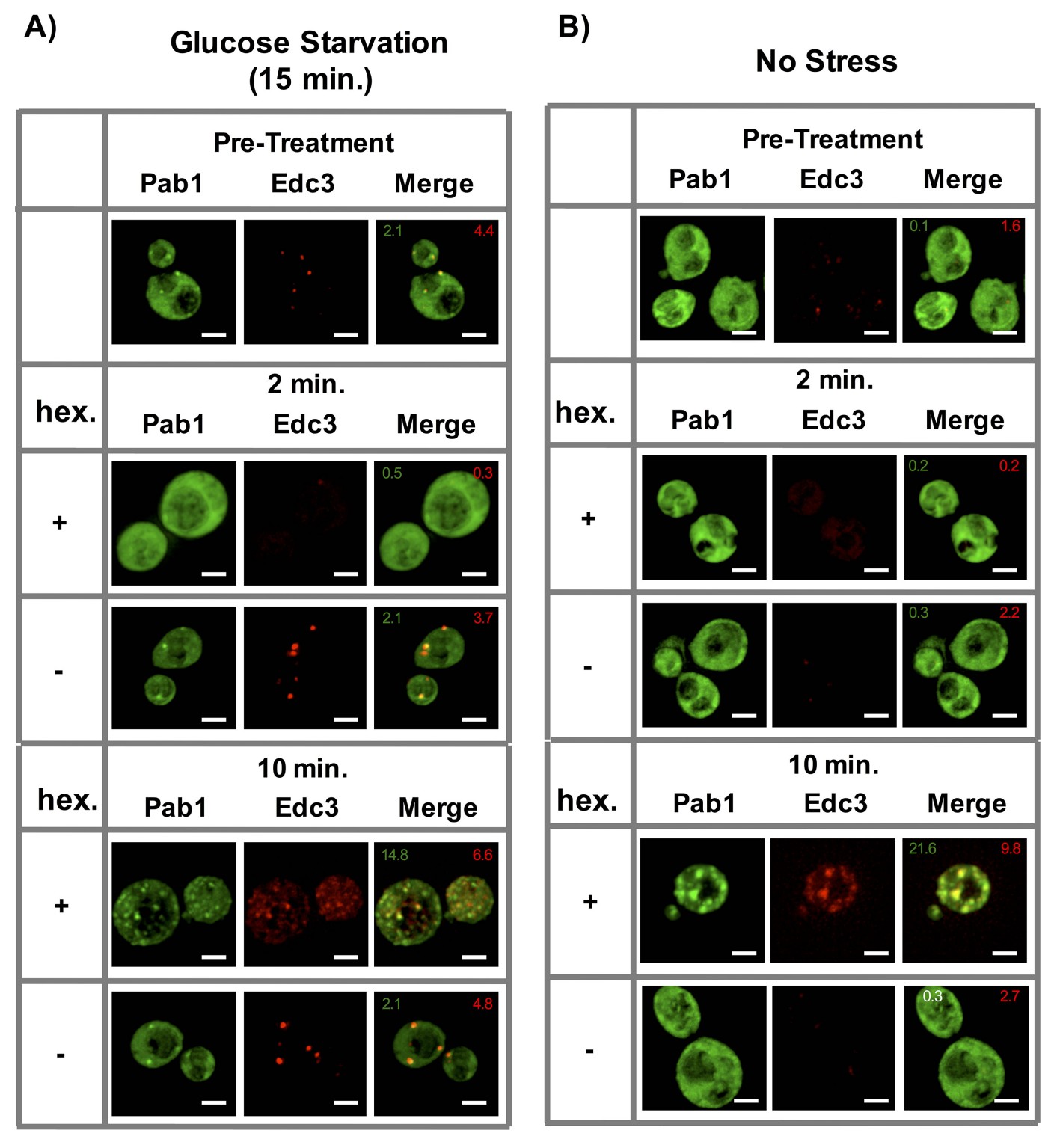

**Figure 8.** Effect of 1,6-Hexanediol on stress granules and P-bodies in yeast. (**A**) Localization of Pab1-GFP and Edc3-mCherry prior to treatment with (Pre-treat) or after 2 min or 10 min of 10 µg/mL Digitonin ±10% 1,6-Hexanediol following 15 min of glucose starvation. All cells continue to be in media lacking glucose throughout the course of the experiment. (**B**) Same as (**A**), after 10 µg/mL Digitonin ±10% 1,6-Hexanediol treatment only. Numbers in green (or white) indicate average number of stress granules/ cell. Numbers in red indicate average number of P-bodies/ cell. All scale bars are 2 µm.

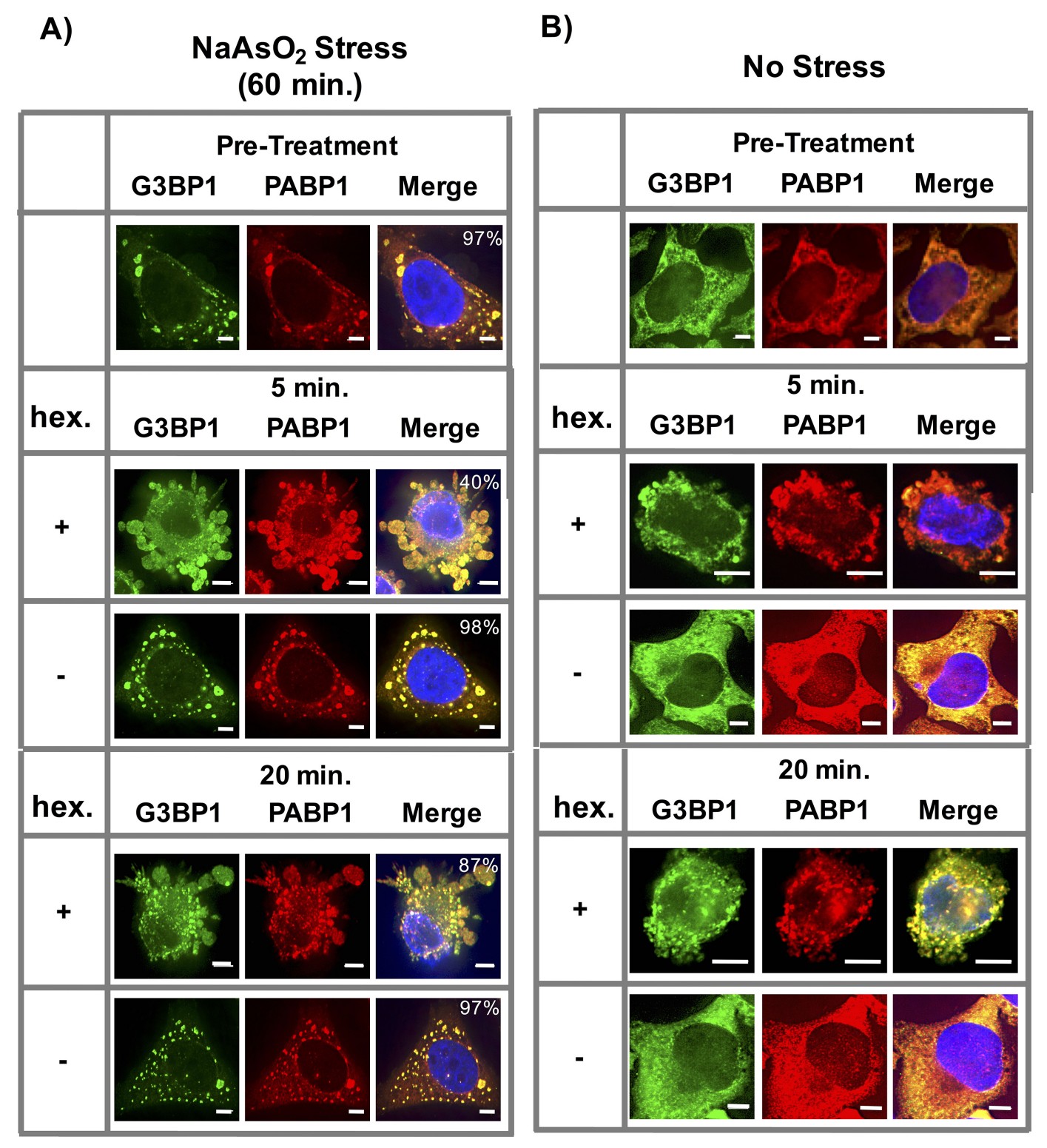

**Figure 9.** 1,6-Hexanediol induces formation of stress granule-like assemblies in HeLa cells. (**A**) Affects of 1,6-Hexanediol on $NaAsO_2$-induced stress granules in HeLa cells. HeLa cells were stressed with $NaAsO_2$ for 60 min. Addition of 3.5% 1,6-Hexanediol in the presence of $NaAsO_2$ was added at indicated time points. For all experiments, HeLa cells were fixed and co-stained for G3BP1 (FITC labeled secondary) and PABP1 (Alexa-647 labeled secondary). Percentages refer to cells with stress granules defined as G3BP1 and PABP1 double positive foci (N=3 per condition, 50 cells/ replicate). (**B**)

*Figure 9 continued on next page*

*Figure 9 continued*

1,6-Hexanediol induces stress granule formation in HeLa cells. HeLa cells were exposed to media containing 3.5% 1,6-Hexanediol for indicated time periods. All scale bars are 5 μm.

larger structures through interactions between the shell structures.

Several lines of evidence argue that stress granule assembly involves the early formation of stable core structures that then assemble into larger stress granules, each of which can contain multiple cores, surrounded by a less dense shell layer (*Figure 12*). First, as soon as stress granules are observed in cells, they are stable in lysates (*Figure 2*). Second, the size distribution of the stable components of stress granules in lysates is similar from 15 to 120 min of stress, arguing that stress granule core formation is not a late step in stress granule assembly (*Figure 4*). Third, as soon as technically possible to assess, the FRAP behavior, as observed by the GFP-G3BP1 mobile fraction, of stress granules remains unchanged from 30 min to 120 min of stress (*Figure 5*). This also argues that stress granules are not undergoing a dramatic change in biochemical state, and is inconsistent with models where stress granules initially consist of a uniform and dynamic LLPS, that then matures into a mature biphasic stress granule. Finally, when examined using super resolution microscopy, we observe granules exhibit a heterogeneous distribution of protein even at early time points (*Figure 3*). Taken together, these observations are consistent with granules assembling from the oligomerization of individual mRNPs into stable core complexes. The oligomerization of individual mRNPs into core structures would be analogous to a liquid-liquid phase separation driven by multivalent interactions, but because the core structures are stable to dilution, could be considered a liquid-solid demixing phase separation. It should be noted we cannot rule out the formal caveat that stress granules initially condense into a weak dynamic assembly, that immediately transitions to a stable assembly, or that stress granule shells and cores form independently as two immiscible liquid phases which later fuse as has been suggested to explain the assembly of nucleolus (*Feric et al., 2016*). We hypothesize that these stable core assemblies could provide a structural platform, due to the high concentration of IDRs on stress granule components to then rapidly phase-separate a dynamic shell that grows as a result of fusion with other small granules and surface exchange with an increasing pool of untranslating mRNPs.

The assembly of stress granule cores as an initial step in assembly solves a conundrum. The issue is that IDRs are thought to interact with each other to drive LLPS often by weak dynamic interactions, proposed to involve Arg-Aromatic interactions in some cases (*Nott et al., 2015*; *Pak et al., 2016*). Since these interactions are limited to few amino acids, one anticipates that any given interaction would be expected to have low specificity, and as such would be expected to interact with many proteins when dispersed in the cytosol. Thus, for such weak and promiscuous interactions to drive *a priori* the assembly of a stress granule would be difficult since the interactions between stress granule components would always be in competition with other cytosolic factors. When the initial step in stress granule assembly is formation of a stable core, which by definition involves relatively stable interactions, the core structure could then provide a high local concentration of IDRs on stress granule components, which could then be envisioned to trigger a local phase separation, possibly what we refer to as the shell phase of a stress granule. This general principle is also seen in the formation of the nuclear pore, wherein the creation of a high local concentration of FG repeat proteins by the structure of the nuclear pore, allows for a phase transition within the nuclear pore itself, even if the interactions between the FG repeats are relatively weak and non-specific (*Frey and Görlich, 2007*; *Hülsmann et al., 2012*; *Schmidt and Görlich, 2016*).

This work provides two observations that suggest a difference between the process of in vitro LLPS driven by IDRs on RNA binding proteins and the formation of stress granules in cells. First, although low temperatures promote LLPS driven by IDRs of RNA binding proteins (*Molliex et al., 2015*; *Nott et al., 2015*), we observed that stress granules formed less efficiently at lower temperatures, despite efficient polysome disassembly (*Figure 6*). Second, although 1,6-Hexanediol can prevent LLPS by IDRs from RNA binding proteins (*Molliex et al, 2015*), we observed that 1,6-Hexanediol effectively induces stress granules in both yeast and mammals (*Figures 8*, *9,* and *10*). These observations suggest that the process of stress granule assembly in cells is not primarily driven by weak, dynamic homo- and heterotypic interactions between IDRs on RNA binding proteins

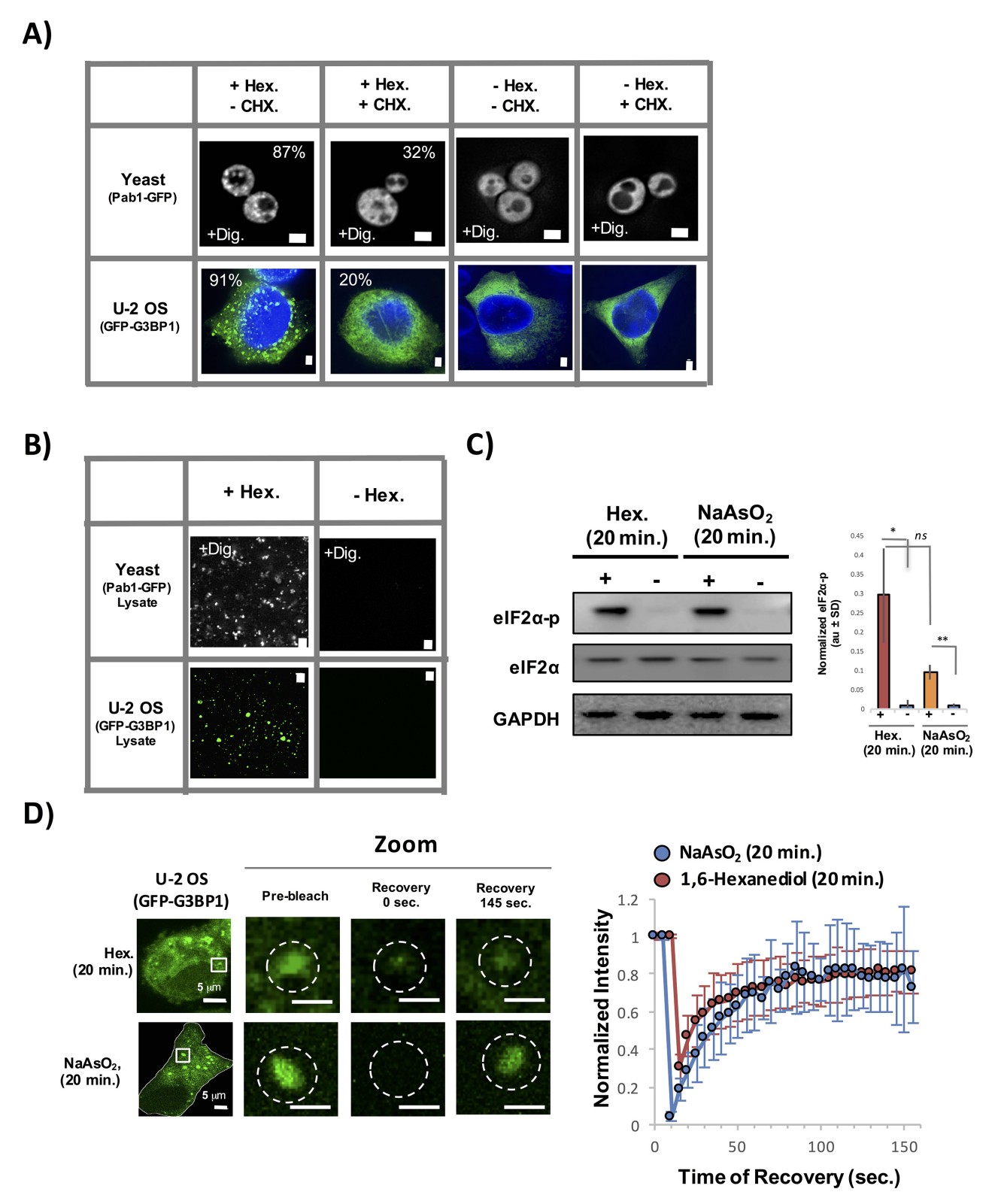

**Figure 10.** Stress granules induced by 1,6-Hexanediol are cyclohexamide sensitive, stable in lysates, and dynamic. (A) 1,6-Hexanediol induced stress granules are cyclohexamide sensitive in yeast and U-2 OS cells. Yeast were grown in the presence of 10% 1,6-Hexanediol and 10 µg/mL Digitonin for 10 min or just 10 µg/mL Digitonin ± cyclohexamide (100 µg/mL). U-2 OS cells were exposed to media containing 3.5% 1,6-Hexanediol for 20 min ± cyclohexamide (10 µg/mL). Percentages refer to cells with stress granules (N=3 per condition). (B) Yeast and U-2 OS 1,6-Hexanediol induced granules

*Figure 10 continued on next page*

*Figure 10 continued*

are stable in lysates. Yeast were grown in presence of 1,6-Hexanediol and 10 µg/mL Digitonin for 10 min or just 10 µg/mL Digitonin prior to lysis. Mammalian cells were grown in the presence or absence of 3.5% 1,6-Hexanediol containing media for 15 min prior to lysis. (**C**) Western blot for eiF2α phosphorylation status following +/− exposure to 1,6-Hexanediol (3.5%) or NaAsO$_2$ (0.5 mM) stress for 20 min. Normalized to eiF2α and GAPDH (N=3, au = arbitary units, SD = standard deviation). (p-value: *<0.01; **<0.001; ns = not significant). (**D**) Granules shown prior to photobleaching, at 0 s, and at 145 s after photobleaching. Cells were treated with either 1,6-Hexanediol or NaAsO$_2$ for 20 min. Graph shows recovery curves as an average of 6 granules ± standard deviation for each respective condition. Abbreviations: Hex., 1,6-Hexanediol; Dig., Digitonin; CHX, cyclohexamide. All scale bars are 2 µm unless otherwise noted.

analogous to the LLPS driven by these protein domains in vitro. However, we hypothesize that weak, dynamic interactions between IDRs could be important in forming the shell structure of stress granules, once the core is nucleated by more stable interactions.

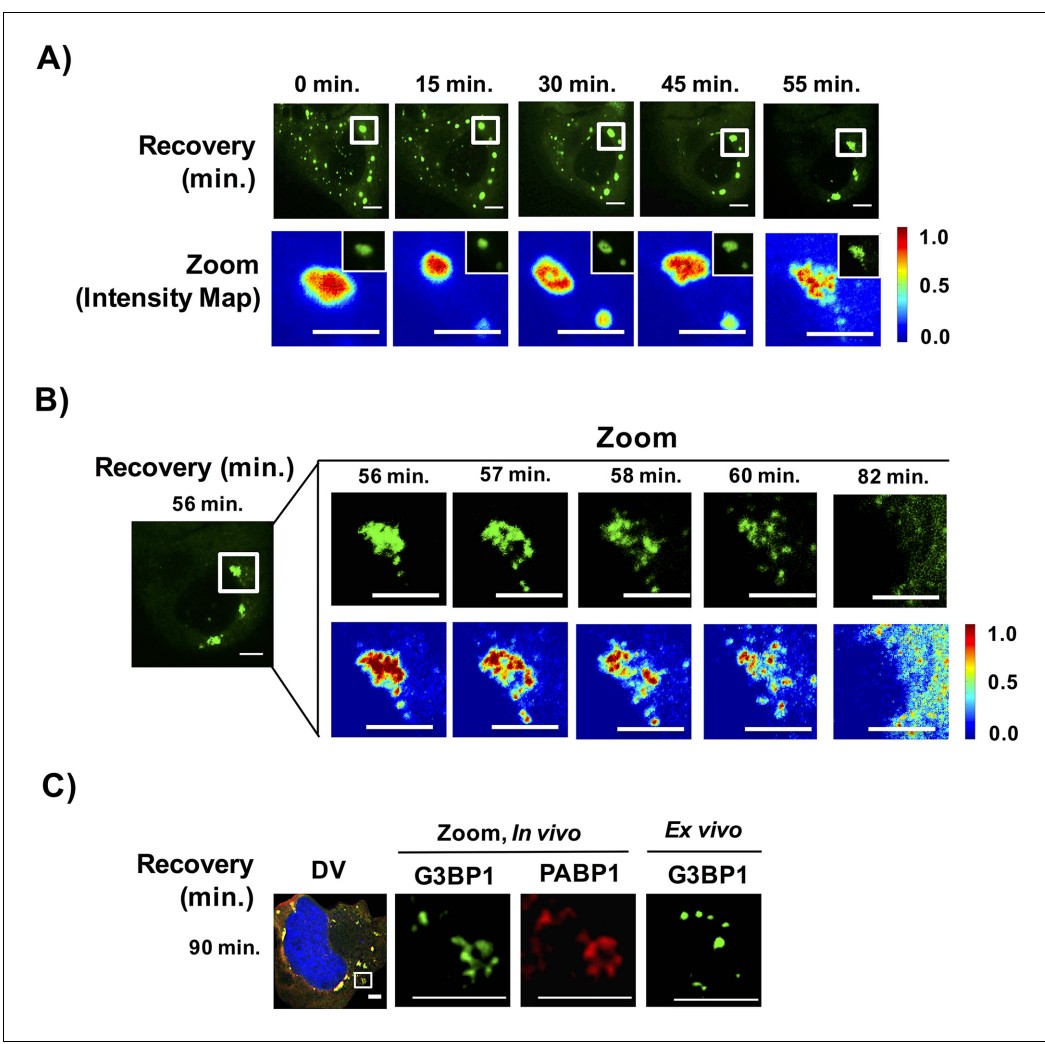

**Figure 11.** Stress granules disassemble through multiple discrete steps. (**A**) Time-lapse imaging of stress granule disassembly in normal media following 60 min of NaAsO$_2$ stress using a 100X objective. Intensity map represents relative gray scale intensity of zoomed inset. (**B**) Same as (**A**) imaged during stress granule fracturing. (**C**) In vivo GFP-G3BP1 stress granules were stained with PABP1 (Alexa-647 labeled secondary) and ex vivo GFP-G3BP1 stress granules cores imaged by deconvolution microscopy (DV) following 90 min of recovery from NaAsO$_2$ stress. All scale bars are 5 µm.

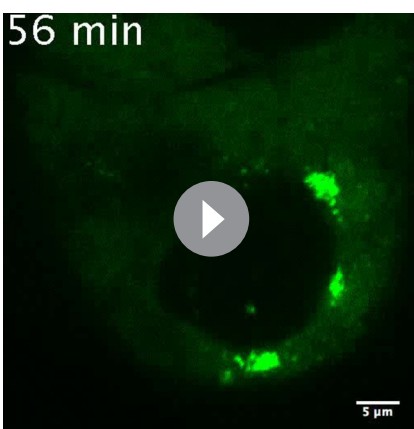

**Video 3.** Stress granule disassembly following NaAsO₂ stress. U-2 OS cells expressing GFP-G3BP1 stressed with NaAsO₂ (0.5 mM) for 60 min and recovered in normal media. Image acquisition begins following 56 min of recovery in normal media. Images were acquired at 20 s intervals on a spinning disk confocal microscope using a 100X objective. Scale bar represents 5 µm. Related to *Figure 11*.

In contrast to our results, 1,6-Hexanediol has been reported to dissolve yeast P-bodies and mammalian stress granules (*Kroschwald et al., 2015*). We do not understand the basis for the difference between our results and those published earlier, although it could be due to differences in cell handling, growth conditions, or analysis. In any case, the sensitivity of a cellular structure to 1,6-Hexanediol does not appear sufficient to conclude the structure forms by LLPS. We suggest this cautionary note for three reasons. First, we observed that some, but not all, cellular structures were sensitive to 1,6-Hexanediol, including well described cytoskeleton elements (*Figure 7*). Second, one should anticipate that, depending on the chemical nature of interactions driving assembly, many different types of assemblies could be sensitive to 1, 6-Hexanediol. Finally, one should also expect LLPS assemblies can form by many types of promiscuous or specific interactions, some of which would be expected to be resistant to 1,6-Hexanediol.

Previous work has suggested that small stress granules that form early in a stress response require microtubules and their associated motors to form larger stress granules (*Chernov et al., 2009*; *Fujimura et al., 2009*; *Ivanov et al., 2003*; *Loschi et al., 2009*). In those experiments, treatment of cells with nocadazole, which depolymerizes microtubules, prevents the fusion of the small initial stress granules into larger assemblies. Based on these results, a reasonable model is that cores are brought together to form larger assemblies by movement on microtubules. Disassembly may involve a reverse process where the microtubule network may facilitate retrograde transport of mRNPs away from a disassembling stress granule.

We observe stress granule disassembly occurs in a reverse process where a less stable shell dissolves initially followed by core disassembly and or clearance by autophagy. This observation has several corollaries. First, upon re-establishment of translation, mRNA is thought to be in rapid equilibrium between the cytosol and stress granules. This exchange appears to influence stress granule structural integrity and may account for titration of select RNAs into translation. The lag in the clearance of granule cores may reflect the requirement of a myriad of ATP-dependent remodeling complexes (e.g. heat shock 70 or p97/VCP AAA-ATPase complexes [*Buchan et al., 2013*; *Walters et al., 2015*]) and could serve as a cytoprotective mechanism to acutely re-nucleate stress granules if the cell re-encounters stress.

In mammalian cells, a biphasic stress granule architecture provides multiple layers of functionality within the larger tunable stress-specific RNA transcriptome. First, a shell in mammalian cells provides a scaffold for dynamic exchange of select RNAs out of the translational pool. This could represent a thermostatic mechanism where select RNAs (possibly important housekeeping genes) could be still translated under stress without overtly outcompeting the translation of stress-specific messages essential for survival under stress. These shell-specific RNAs would be expected to be primed for reentry into translation during recovery and may also represent a metastable structural scaffold that fractures during stress granule disassembly. Meanwhile, further compartmentalization into less dynamic cores could provide an additional layer of sequestration of select messages. This could possibly serve as a biochemical sorting mechanism where 'fit' mRNPs could be remodeled under stress by concentrating similar RNPs, while 'unfit' mRNPs could be sorted for decay or not remodeled to reenter the translational pool.

In degenerative disease, this multistep process might become altered leading to aberrant formation of granules. Hyper-nucleation or inefficient clearance of stress granule cores may facilitate a

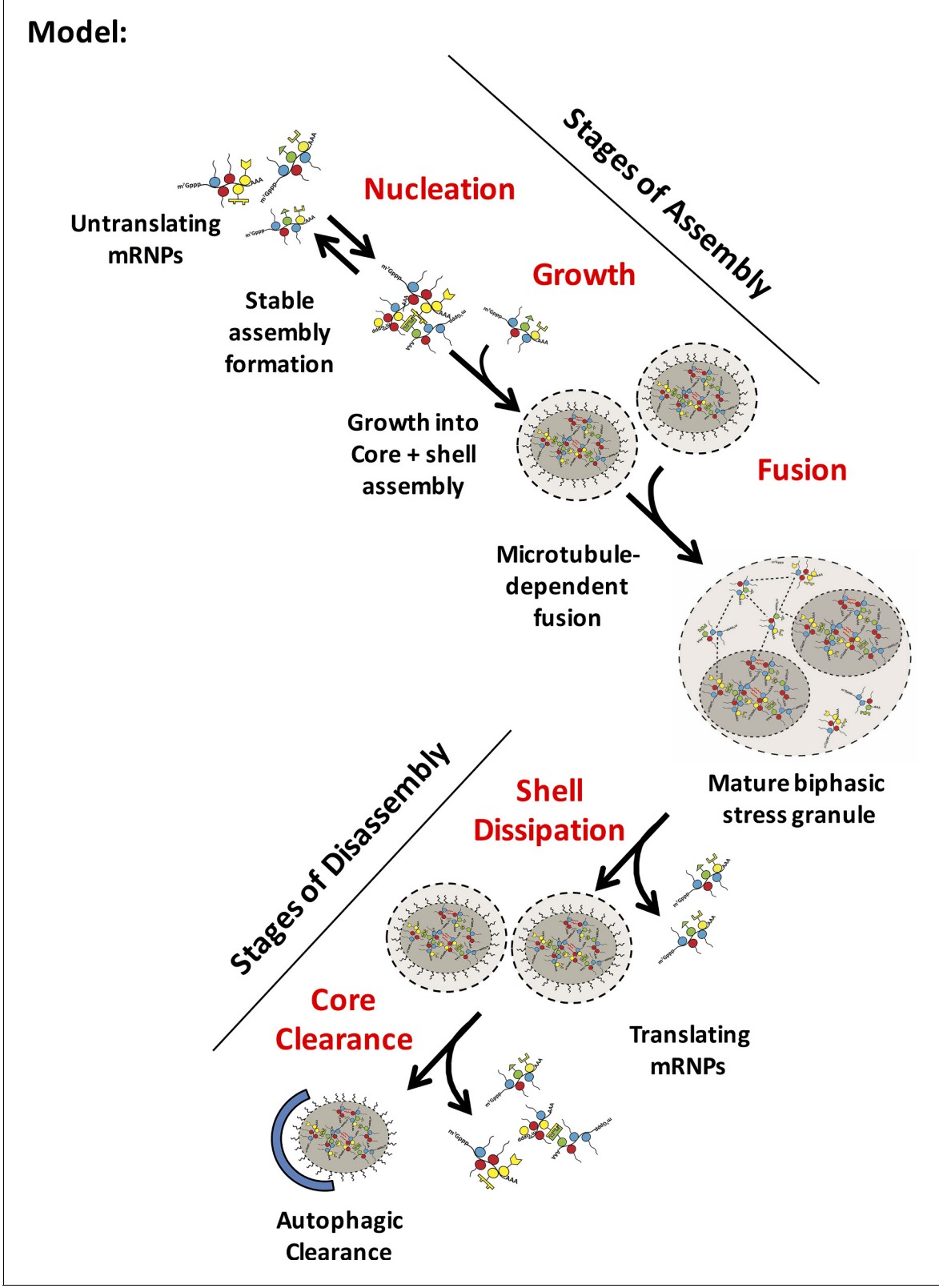

**Figure 12.** Model for distinct stages of stress granule assembly and disassembly Possible steps in granule assembly and disassembly are shown. Untranslating mRNPs nucleate an early stable mRNP (core) complex and grow to rapidly include an early core/ shell (biphasic) assembly through an ATP-dependent, microtubule-independent process. Fusion of biphasic stress granules forms a larger higher order mature assembly in part by a microtubule-dependent process. Disassembly is likely to occur through shell dissipation by exchange of weakly associated granule (shell) mRNPs into a

*Figure 12 continued on next page*

*Figure 12 continued*

recovering translational mRNP pool. More stable core assemblies may then be disassembled by ATP-requiring remodeling complexes or autophagy. Dashed lines between mRNPs represent weak physical interactions in phase-separated shell (light gray). Red wavy lines represent strong interactions between IDRs in stable cores (darker gray).

pathological transition by lowering the free energy landscape to seed an β-amyloid aggregate by providing a continued platform for nucleation.

## Materials and methods

### Yeast and mammalian cell growth conditions

All GFP-tagged strains shown in *Figure 7* (Sac6-GFP, Tub1-GFP, Nsp1-GFP, Get1-GFP and Atp1-GFP) were taken from the yeast GFP collection. These strains were grown in minimal media supplemented with a complete set of amino acids and 2% Dextrose at 30°C. For the experiments presented in *Figure 8*, BY4741 yeast was transformed with a single plasmid expressing Pab1-GFP and Edc3-mCherry (pRP1657). These strains were grown in –Ura media with 2% Dextrose at 30°C. Experiments in *Figure 10A* were done with 4741 transformed with a plasmid expressing Pab1-GFP (pRP 1363). Human osteosarcoma U-2 OS (expressing GFP-G3BP1, mRFP-DCP1a) (*Kedersha et al., 2008*; *Ohn et al., 2008*) and HeLa cells, maintained in DMEM, High Glucose, GlutaMAX with 10% fetal bovine serum, 1% penicillin/streptomycin, and 1 mM sodium pyruvate at 37°C/5% $CO_2$ were used for all biochemical experiments, stress granule purification, and imaging experiments.

### Stressing cells

Yeast: $NaN_3$ stress: Cells were treated with 0.5% $NaN_3$ for 30 min at 30°C (unless otherwise stated). Mammalian: $NaAsO_2$ stress: Cells were treated with 0.5 mM $NaAsO_2$ at 37°C/5% $CO_2$ (unless otherwise stated). Thapsigargin stress: Cells were treated with 100 nM Thapsigargin in DMSO for 1 hr at 37°C/5% $CO_2$. Osmotic stress: Cells were treated with 375 µM Sorbitol at 37°C/5% $CO_2$ for 1 hr.

### Analysis of yeast and mammalian stress granule cores

A 'granule-enriched' fraction was prepared from yeast and mammalian cells as previously described (*Jain et al., 2016*). For yeast granule isolation, 50 mL of yeast were grown. For mammalian granule isolation, stressed U-2 OS cells were collected from 15 cm plates, pelleted and snap frozen in liquid nitrogen. Cell pellets (re-suspended in lysis buffer) were used to prepare granule enriched fractions. In lysate immunofluorescence and oligo-(dT) staining of stress granule cores was performed as previously described (*Jain et al., 2016*). 1,6-Hexanediol granules were isolated from U-2 OS cells grown in the presence of 3.5% 1,6-Hexanediol for 15 min.

### Nanosight particle size analysis

Stress granule core enriched fractions were collected at respective time points and diluted in lysis buffer (1:2). Particles were analyzed using the NanoSight Nanoparticle Tracking Analysis system NS300 (Malvern) with syringe pump, a 488 nm laser, and a sCMOS camera. Five videos of 60 s were collected for each sample using the 488 nm laser and analyzed by NTA 3.0 software. Maximal concentration of particles (particles/ nm) was used for normalization.

### FRAP analysis of U-2 OS GFP-G3BP1 cells

U-2 OS cells were cultured on glass bottom 35 mm dishes. FRAP experiments were performed at indicated times as previously described (*Jain et al., 2016*).

### Digitonin, cyclohexamide and 1,6-Hexanediol treatment in yeast

Cells were grown to an $OD_{600}$ of 0.4–0.7 in minimal media supplemented with the appropriate amino acids, at 30°C. These cells were then imaged for the 'No Drug' or 'Pre-treat' controls. These cultures were then pelleted at 3220Xg for 1 min, washed once with medium containing 10 µg/ ml

Digitonin ± 10% 1,6-Hexanediol (similar to the concentrations used in *Kroschwald et al., 2015*), then resuspended in the same medium and returned to a shaker at 30°C. Cells were then imaged after 2 min, 10 min, 20 min and 30 min. For glucose starvation experiments, cells at log phase were pelleted at 3220Xg for 1 min, washed once in media without Dextrose, then resuspended in media without Dextrose and returned to a shaker at 30°C. After 15 min, these cells were then pelleted again at 3220Xg for 1 min and washed once and then resuspended in dextrose free media containing 10 µg/ ml Digitonin ± 10% 1,6-Hexanediol. These cells were then returned to a shaker at 30°C, and imaged after 2, 10, 20 and 30 min. All cyclohexamide treatments were performed at a final concentration of 100 µg/mL added to media at the same time as Digitonin ± 1,6-Hexanediol.

## 1,6-Hexanediol Treatment in HeLa and U-2 OS cells

For treatment with 1,6-Hexanediol, complete media (DMEM supplemented with 10% FBS, 1% penicillin/streptomycin) was prepared containing 3.5% 1,6-Hexanediol. Media was exchanged and replaced with 3.5% 1,6-Hexanediol or complete media for indicated time periods. HeLa cells were stressed with $NaAsO_2$ (0.5 mM) for 1 hr. Media was exchanged and replaced with 3.5% 1,6-Hexanediol and $NaAsO_2$ (0.5 mM) or $NaAsO_2$ (0.5 mM) only containing media at indicated time points. For all experiments, HeLa cells were fixed (4% paraformaldehyde) and co-stained for PABP1 (detected by Cy5 labeled secondary) and G3BP1 (detected by FITC labeled secondary). The same concentrations for staining were used in all experiments for both primary (1:200) and secondary (1:400) antibodies. For U-2 OS experiments, all experiments were conducted with 3.5% 1,6-Hexanediol, cyclohexamide (10 µg/mL), and $NaAsO_2$ (0.5 mM) prepared in complete media.

## Cellular viability measurements

Cellular viability was measured using RealTime-Glo MT Cell Viability Assay (Promega) according to manufacture's protocol. HeLa cells were exposed to complete media containing 3.5% 1,6-Hexanediol for indicated time periods. Following exposure, normal media was exchanged and cells were incubated in presence of Realtime-Glo substrates for 1 hr or 14 hrs at 37°C. Luminescence measurements were performed using a Victor[3] plate reader (PerkinElmer; Waltham, MA).

## Microscopy and image analysis

All yeast and mammalian images were acquired using a DeltaVision Elite microscope with a 100X objective using a PCO Edge sCMOS camera unless otherwise stated. $\geq$3 images were taken for each experiment comprising of 8 Z-sections each. Stress granule cores were visualized using Deltavision. All images were analyzed using ImageJ. In *Figure 2*, three independent experiments were performed for each time point. Granule core numbers were calculated using Deltavision, maximal image projections were determined using ImageJ, and thresholding was kept constant across all replicates and time points. Plotted are mean GFP-G3BP1 postiive particles (± standard deviation) normalized to particles at time zero. The x-axis was zeroed to 50 to allow for easier comparison within the graph. In *Figure 9*, percent HeLa cells (N=3, 50 cells/ replicate) with stress granules (G3BP1 and PABP1 double positive foci) were calculated for each condition.

Live cell imaging was performed using a Nikon Spinning Disk Confocal microscope outfitted with an environmental chamber with $O_2$, $CO_2$, temperature, and humidity control. All images were acquired using a 100X objective with a 2x Andor Ultra 888 EMCCD camera. For temperature experiments, U-2 OS cells grown at 37°C in 35 mm dishes were transferred to environmental chamber at respective temperatures and allowed to equilibrate to respective temperatures for 10 min prior to image acquisition. For disassembly experiments, cells were washed twice in normal media and permitted to recover during image acquisition. All images were acquired by exciting for 200 ms using a 488 nm laser at a gain of 200 (unless otherwise stated). In *Figure 2*, GFP-G3BP1 foci were determined from thresholded (Otsu) images acquired at 20 s time intervals and plotted as a percentage of peak foci detected for 9 total cells acquired from three independent experiments. In *Figure 7*, bright field movie was acquired using a Nikon spinning disk microscope using a 20X air objective (HBO arc lamp). Time and event stamps and scale bars were added using ImageJ.

SIM (Structured Illumination Microscopy) was performed using the Nikon N-SIM microscope system run in 3D SIM mode with a Nikon 100X objective at a 1.49NA. Images were acquired using an Andor iXon DU897 EM-CCD camera. GFP-G3BP1 was visualized by exciting the sample using a

488 nm laser at a gain of 300. Samples were mounted using ProLong Diamond Mounting media incubated overnight at room temperature. Sample was excited for 600 ms/ image. Image reconstruction was performed in Nikon NIS Elements.

## Acknowledgements

We thank Denise Muhlrad, Robert Walters, Carolyn Decker, Sarah Mitchell, David Protter, and the Parker Lab for helpful discussions and feedback on the manuscript. The imaging work was performed at the CU Light Microscopy Core Facility and the BioFrontiers Institute Advanced Light Microscopy Core. Laser scanning confocal microscopy was performed on a Nikon A1R microscope acquired by the generous support of the NIST-CU Cooperative Agreement award number 70NANB15H226. Spinning disc confocal microscopy was performed on a Nikon Ti-E microscope acquired by the generous support of Professor Tom Cech, Professor Roy Parker, and the Howard Hughes Medical Institute. SIM imaging was performed on a Nikon N-SIM structured illumination super-resolution made possible by equipment supplements to R01 GM79097 (D Xue) and P01 GM105537 (M Winey).

## Additional information

### Funding

| Funder | Grant reference number | Author |
|---|---|---|
| Howard Hughes Medical Institute | | Roy Parker |
| National Institute of Neurological Disorders and Stroke | F30N2093682 | Joshua R Wheeler |
| National Institute of General Medical Sciences | GM045443 | Roy Parker |

The funders had no role in study design, data collection and interpretation, or the decision to submit the work for publication.

### Author contributions

JRW, TM, SJ, Conception and design, Acquisition of data, Analysis and interpretation of data, Drafting or revising the article; RA, Investigation, Acquisition of data; RP, Conception and design, Analysis and interpretation of data, Drafting or revising the article

### Author ORCIDs

Joshua R Wheeler, http://orcid.org/0000-0002-7315-8269
Roy Parker, http://orcid.org/0000-0002-8412-4152

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
