## [Decision Letter]

Thank you for submitting your article "Distinct stages in stress granule assembly and disassembly" for consideration by *eLife*. Your article has been reviewed by three peer reviewers, and the evaluation has been overseen by Timothy Nilsen as the Reviewing Editor and James Manley as the Senior Editor. The reviewers have opted to remain anonymous.

The reviewers have discussed the reviews with one another and the Reviewing Editor has drafted this decision to help you prepare a revised submission.

We have now received comments on your manuscript from three expert referees. As you will see, they were generally positive about the work but several issues will need to be dealt with via revision. Specifically points 2, 3 and 4 raised by Reviewer 2 as well as his/her minor points should be addressed thoroughly. In addition all of the referees felt that some conclusions were overstated or not rigorously supported by experiments. Please soften these statements appropriately.

Reviewer #1:

In this manuscript the authors investigated the steps/mechanisms modulating the assembly/disassembly of Stress Granules during arsenite-induced stress. The authors performed a series of experiments which included: (1) verifying the kinetic of SG assembly in cells exposed to arsenite and overexpressing GFP-G3BP1 (2) examining the ultrastructure of these SGs using super-resolution microscopy (3) assessing the dynamics and core size dimensions at different time during SG formation (4) assessing their assembly at lower temperature in order to determine how SGs form. By performing these experiments the authors concluded that the assembly of SGs begins with the formation of a core and the subsequent assembly of the shell.

The disassembly also proceeds in a stepwise logical manner starting with the removal of the shell and the clearance of the core.

This is a very interesting manuscript which attempts to understand the mechanisms behind the assembly of SGs. The data provided are convincing, appropriate and support the conclusions set forth by the authors. There are, as such, no major concern with the work.

Reviewer #2:

In this manuscript by Wheeler et.al. "Distinct stages in stress granule assembly and disassembly", the nature of stress granule heterogeneity and assembly dynamics are interrogated. Previous work by these authors had revealed that stress granules have a structure defined by a more stable "core", surrounded by a more dynamic "shell". Here the authors utilize light microscopy approaches, combined with chemical and physical (i.e. temperature) perturbations, to test two different models: 1) the core is formed first followed by a shell, or 2) the whole structure assembles and then a more mature core nulceates within. The authors conclude that the first model is correct. The conclusion of initial core assembly followed by dynamic shell formation is based on some findings indicating stable cores are present in stress granule from the early stage of assembly, and that the assembly of a liquid-like shell is not required for stress granule assembly. Overall, this study has some interesting observations but the conclusions are not strongly supported. In particular:

1) No evidence has been presented to show that there is no shell in any conditions examined. In order for the claim that stress granule assembly proceeds with the initial core formation to be valid, showing the presence of core at the early time point of assembly is not enough but showing the absence of shell in the initial stress granule is necessary. FRAP data in Figure 5 clearly indicate that most of G3BP in stress granule shows high mobility, consistent with liquid-like shells, 30 min after NaAsO2 treatment, which is an early time point of assembly by their definition. If shells are present in stress granule from the earliest time point they can study, it is hard to justify their argument that stress granule assembly is initiated by stable core formation.

2) Based on the fact that treatment of yeast and mammalian cells with 1,6-Hexanediol lead to formation of stress granules, the authors draw a conclusion that "the assembly of a LLPS shell is not necessary for stress granule assembly" (Results section, subsection “Stress granule disassemble through multiple discrete steps”, Figure 8 and Figure 9). However, no evidence has been provided for the material state of 1,6-Hexanediol induced stress granules. This claim would require FRAP measurements on these stress granules, to examine if they are indeed mostly stable cores lacking liquid-like shells.

Related to this, it is necessary to discuss the potential origin of discrepancies in experimental results for 1,6-Hexanediol treatments between this work and Kroschwald et al..

3) In Figure 2, there are already foci in cell lysate at time 0. Does this indicate the presence of stable cores without any stress? How do these pre-existing foci affect the size distribution measured in Nanosight experiments (Figure 4)? How does the size distribution look in the earlier time point, 10 and 20 min? This is relevant since by 30 min after NaAsO2 treatment the growth phase of stress granule assembly is almost finished (based on Figure 1).

4) Figure 6, what time point after NaAsO2 treatment is used for this western blot? How about at lower temperatures, like 27C? It looks like there is higher phosphorylation at 37C. This data is important to fully rule out the possibility of slower eIF2α phosphorylation at lower temperatures as an origin for delayed SG assembly.

5) A recent paper by Feric et.al. Cell 165(7)2016 describes liquid immiscibility of a as an organizing principle for the nucleolus. The model presented in that paper is one of two liquid droplets that do not mix with one another, with the core liquid prone to gelation over time- this seems to be a third model not considered in the current manuscript. Do the data rule out a model like this for stress granules?

Reviewer #3:

Stress granules are cytoplasmic assemblies consisting of stable cores surrounded by a dynamic shell. Some stress granule proteins are able to undergo liquid-liquid phase separation (LLPS) in vitro, raising the possibility that LLPS may play a role in stress granule assembly. This model has gained some support recently since it has been demonstrated that liquid droplets can mature into more stable structures over time. How LLPS initially provides the specificity needed to recruit stress granule components and exclude other cytoplasmic proteins, however, is not known. An alternative model is that stress granule assembly is initiated by the formation of stable complexes through conventional protein/protein interactions. These cores create localized foci of high concentration of RNA-binding proteins, which in turn template local phase separation to create the more dynamic shell. In the present study, the authors present evidence in support of core-first/shell-later model using a time course analysis of stress granule assembly. They demonstrate that stable cores appear in cell lysates within minutes of stress induction and that stress granules have a non-homogeneous appearance immediately upon assembly. The authors also show that exposure to low temperature, which should enhance LLPS, interferes with stress granule formation. Furthermore, they show that treatment with hexanediol, which has been used to dissolve weak assemblies in cells, affects too many cellular structures to be diagnostic and in fact induces stress granules.

Although skeptics may argue that these studies are too descriptive to provide insights into mechanism (formation of small liquid droplets may be hard to visualize), the observations presented are certainly provocative and provide a reasonable model alternative for stress granule assembly. My only concern is the description of granule disassembly, which the authors describe as "shell dissipation" and "core fracturing". This seems to be an over-interpretation of the images shown in Figure 10, given their low resolution. But I agree with the authors that these images do not appear to be consistent with a simple phase transition. It may be helpful to point out to the readers the features (filaments?) that appear at the edges of the stress granules during disassembly (Figure 10, 60 min timepoint).

In the Discussion, I suggest that the authors distinguish which aspects of their model is supported by data and which are more speculative or only supported by in vitro experiments. For example, what is the evidence that the dynamic shell is formed by phase separation and causes granules to fuse with one another?

In summary, this study provides several interesting observations that challenge a popular model for granule assembly. This study will be of interest to cell biologists and to the growing field of biophysicists that study phase separation.

---

## [Author Response]

*1) No evidence has been presented to show that there is no shell in any conditions examined. In order for the claim that stress granule assembly proceeds with the initial core formation to be valid, showing the presence of core at the early time point of assembly is not enough but showing the absence of shell in the initial stress granule is necessary. FRAP data in Figure 5 clearly indicate that most of G3BP in stress granule shows high mobility, consistent with liquid-like shells, 30 min after NaAsO2 treatment, which is an early time point of assembly by their definition. If shells are present in stress granule from the earliest time point they can study, it is hard to justify their argument that stress granule assembly is initiated by stable core formation.*

Our model suggests stress granule core formation is an early event in stress granule assembly. The concern presented by the reviewer is whether we can also rule out if shell formation is also an early event, perhaps concomitant with core assembly, based on the data presented. We agree that our data does not conclusively rule out the possibility of shell formation as also being an early event. However, we point out that two perturbations expected to either decrease LLPS (1,6-Hexanediol treatment) or increase LLPS (lower temperatures), provide evidence that a 1,6-Hexanediol sensitive, low temperature induced LLPS is not a pre-requisite for stress granule formation. Specifically, the treatment thought to decrease LLPS (1,6-Hexanediol) actually triggers robust stress granule formation in yeast and mammals, and the treatment thought to increase LLPS (lower temperatures) actually inhibits stress granule formation. We agree with the reviewer that we cannot rule out that cores and shells form essentially at the same time, and we have re-written the text to reflect this possibility.

*2) Based on the fact that treatment of yeast and mammalian cells with 1,6-Hexanediol lead to formation of stress granules, the authors draw a conclusion that "the assembly of a LLPS shell is not necessary for stress granule assembly" (Results section, subsection “Stress granule disassemble through multiple discrete steps”, Figure 8 and Figure 9). However, no evidence has been provided for the material state of 1,6-Hexanediol induced stress granules. This claim would require FRAP measurements on these stress granules, to examine if they are indeed mostly stable cores lacking liquid-like shells.*

Two issues are raised by the reviewer: first, whether the assemblies induced by 1,6-Hexanediol treatment are in fact bona fide stress granules; and two, whether the 1,6-Hexanediol-induced assemblies are more ‘core-like’ (although this is not a specific conclusion we intended to make), To address these issues, we examined the properties of 1,6-Hexanediol-induced assemblies to see how they compare to normal stress granules.

We have performed several additional experiments (now shown as Figure 10) that collectively argue 1,6-Hexanediol induces stress granule formation in a manner similar to other stresses. First, as previously shown, 1,6-Hexanediol induces the formation of assemblies that contain known stress granule proteins in both yeast and mammalian cells (HeLa and U-2 OS cells). Second, we now show 1,6-Hexanediol induced assemblies are sensitive to cycloheximide treatment, a drug known to block stress granule assembly, in both yeast and mammalian cells. Third, we now show 1,6-Hexanediol-induced assemblies from both yeast and mammals are stable in lysates suggesting that the material properties of these assemblies contain a relatively stable set of interactions similar to normal stress granules. Fourth, we show that eiF2α phosphorylation is induced to a similar extent by 1,6-Hexanediol compared to arsenite, consistent with 1,6-Hexanediol inducing stress granules by inhibiting translation initiation. Finally, as requested, we provide FRAP of 1,6-Hexanediol-induced granules in mammalian cells which indicates that 1,6-Hexanediol induced stress granules behave similar to normal stress granules. These observations lead us to conclude that 1,6-Hexanediol induces stress granules in yeast and mammalian cells.

These experiments can be interpreted in two manners: 1) that 1,6-Hexanediol disrupts LLPS in cells, and, since 1,6-Hexanediol induces stress granules indistinguishable from arsenite induced stress granules, therefore no LLPS is required for stress granule formation, or 2) that 1,6-Hexanediol is a very non-specific probe and interpretations of its effect should be carefully considered. We have re-written the discussion to clarify these interpretations.

*Related to this, it is necessary to discuss the potential origin of discrepancies in experimental results for 1,6-Hexanediol treatments between this work and Kroschwald et al.*

We have attempted in the discussion to explain the origins of discrepancy between our results and those presented by Kroschwald et al.

*3) In Figure 2, there are already foci in cell lysate at time 0. Does this indicate the presence of stable cores without any stress? How do these pre-existing foci affect the size distribution measured in Nanosight experiments (Figure 4)? How does the size distribution look in the earlier time point, 10 and 20 min? This is relevant since by 30 min after NaAsO2 treatment the growth phase of stress granule assembly is almost finished (based on Figure 1).*

The reviewer’s concern is whether our experiments sufficiently describe the early process of stress granule formation events to suggest cores as an early event and not as a result of rapid stress granule maturation. As the reviewer points out, our data reveals GFP-positive foci in lysates at time zero. This is because we always observe a few cells in even unstressed cultures that have stress granules. The key point of this experiment is that as stress granules are induced in cells, we observe an increasing amount of stress granule cores stable in lysates, even at early times. Thus, we feel confident in concluding that, at least as soon as we can detect stress granules in cells, stress granule cores are stable in lysates.

The reviewer also asks us to examine how the size of cores changes over very early times (10-20 minutes of stress) and whether cores are growing during this time window. To directly test whether core size increases during the ‘growth’ phase, we have examined the size using nanoparticle tracking analysis of stress granule cores at 15 minutes of arsenite stress. We observe a similar size distribution even at 15 minutes of arsenite stress, as compared to 30’, suggesting that granule cores even at early time points do not appreciably increase in size as compared to the later time points assayed. Together, we believe this result strengthens our conclusion that stress granule cores may represent an early stable compartment that serves as a platform for the accretion of substrate into the larger shell.

*4) Figure 6, what time point after NaAsO2 treatment is used for this western blot? How about at lower temperatures, like 27C? It looks like there is higher phosphorylation at 37C. This data is important to fully rule out the possibility of slower eIF2α phosphorylation at lower temperatures as an origin for delayed SG assembly.*

The main issue being addressed here is similar to that of reviewer #1, and is whether translation repression is truly temperature independent, which is important to demonstrate in order to infer that stress granule assembly per se is what is affected by temperature. As detailed in comments to reviewer #1 above, we have addressed this by quantifying the eIF2α phosphorylation experiments and by adding polysome analysis of arsenite repression at different temperatures. Together, we believe this strengthens our claim that stress granule formation is delayed at lower temperatures despite being in conditions, sufficient substrate and lower temperatures, predicted to enhance phase separation (i.e. granule formation).

In addition, as requested by the reviewer, we provide the time point analyzed in the figure (20’) and in the figure legend.

*5) A recent paper by Feric et.al. Cell 165(7)2016 describes liquid immiscibility of a as an organizing principle for the nucleolus. The model presented in that paper is one of two liquid droplets that do not mix with one another, with the core liquid prone to gelation over time- this seems to be a third model not considered in the current manuscript. Do the data rule out a model like this for stress granules?*

We have now referenced this paper in our discussion. We should point out that this paper was published after we submitted our paper.

*Reviewer #3:*

*My only concern is the description of granule disassembly, which the authors describe as "shell dissipation" and "core fracturing". This seems to be an over-interpretation of the images shown in Figure 10, given their low resolution. But I agree with the authors that these images do not appear to be consistent with a simple phase transition. It may be helpful to point out to the readers the features (filaments?) that appear at the edges of the stress granules during disassembly (Figure 10, 60 min timepoint).*

We thank the reviewer for their careful assessment of our microscopy data. We have added a sentence noting the appearance of features at the edges of stress granules during disassembly, however since we do not know their nature we have not discussed their possible roles in the process.